# Vision-related convergent gene losses reveal *SERPINE3*'s unknown role in the eye

Henrike Indrischek[1,2,3,4,5,6], Juliane Hammer[7], Anja Machate[7], Nikolai Hecker[2,3,4†], Bogdan Kirilenko[1,2,3,4,5,6], Juliana Roscito[2,3,4‡], Stefan Hans[7], Caren Norden[2§], Michael Brand[7*], Michael Hiller[1,2,3,4,5,6*]

[1]Senckenberg Research Institute, Frankfurt, Germany; [2]Max Planck Institute of Molecular Cell Biology and Genetics, Dresden, Germany; [3]Max Planck Institute for the Physics of Complex Systems, Dresden, Germany; [4]Center for Systems Biology Dresden, Dresden, Germany; [5]LOEWE Centre for Translational Biodiversity Genomics, Frankfurt, Germany; [6]Faculty of Biosciences, Goethe-University, Frankfurt, Germany; [7]Center for Regenerative Therapies Dresden, TU Dresden, Dresden, Germany

**\*For correspondence:**
michael.brand@tu-dresden.de
(MB);
michael.hiller@senckenberg.de
(MH)

**Present address:** †Center for Brain & Disease Research, Department of Human Genetics, Leuven, Belgium; ‡DRESDEN-concept Genome Center, DFG NGS Competence Center, c/o Center for Molecular and Cellular Bioengineering, Dresden, Germany; §Instituto Gulbenkian de Ciência, Oeiras, Portugal

**Abstract** Despite decades of research, knowledge about the genes that are important for development and function of the mammalian eye and are involved in human eye disorders remains incomplete. During mammalian evolution, mammals that naturally exhibit poor vision or regressive eye phenotypes have independently lost many eye-related genes. This provides an opportunity to predict novel eye-related genes based on specific evolutionary gene loss signatures. Building on these observations, we performed a genome-wide screen across 49 mammals for functionally uncharacterized genes that are preferentially lost in species exhibiting lower visual acuity values. The screen uncovered several genes, including *SERPINE3*, a putative serine proteinase inhibitor. A detailed investigation of 381 additional mammals revealed that *SERPINE3* is independently lost in 18 lineages that typically do not primarily rely on vision, predicting a vision-related function for this gene. To test this, we show that *SERPINE3* has the highest expression in eyes of zebrafish and mouse. In the zebrafish retina, *serpine3* is expressed in Müller glia cells, a cell type essential for survival and maintenance of the retina. A CRISPR-mediated knockout of *serpine3* in zebrafish resulted in alterations in eye shape and defects in retinal layering. Furthermore, two human polymorphisms that are in linkage with *SERPINE3* are associated with eye-related traits. Together, these results suggest that *SERPINE3* has a role in vertebrate eyes. More generally, by integrating comparative genomics with experiments in model organisms, we show that screens for specific phenotype-associated gene signatures can predict functions of uncharacterized genes.

## Editor's evaluation

The authors use a comparative genomics approach to predict gene function, in particular genes that have a role in eye development. After identifying the convergent loss of SERPINE3 with vision loss across mammals, the authors confirmed its involvement in eye development by characterizing zebrafish knockouts. This work highlights the power of comparative genomics to generate hypotheses that can be experimentally validated. This work is relevant to a broad audience interested in evolution and adaptation as well as for those studying eye development and eye pathologies.

## Introduction

Disorders affecting eyes range from subtle vision impairment to blindness and are among the most prevalent diseases in the human population (*Sheffield and Stone, 2011*). For example, an estimated 76 million people worldwide suffer from glaucoma (*Allison et al., 2020*), a disease involving optic nerve damage, and about 1.8 million people are blind due to age-related macular degeneration, a degenerative disease of the central retina affecting retinal pigment epithelium (RPE) and photoreceptor cells (*Mitchell et al., 2018*; *GBD 2019 Blindness and Vision Impairment Collaborators and Vision Loss Expert Group of the Global Burden of Disease Study, 2021*).

Extensive research in the past decades identified many protein-coding genes with crucial roles in development and maintenance of different tissues and cell types in the eye as well as numerous genes that are associated with genetic eye disorders (*Sheffield and Stone, 2011*; *Moore et al., 2018*). For example, the RetNet database (*Daiger et al., 2022*) lists 271 genes associated with heritable retinal diseases. Even though the eye probably represents one of the best studied organs, our knowledge of the genes underlying eye diseases and disorders is still incomplete. For example, linkage analysis in patients with cataract, microcornea, microphthalmia, and iris coloboma identified new genomic loci linked to the diseases; however, the disease-causing genes have remained elusive (*Wang et al., 2007*; *Sabir et al., 2010*; *Abouzeid et al., 2009*). Similarly, the Cat-Map database (*Shiels et al., 2010*) lists several additional cataract-associated loci where the underlying disease-causing gene has not been identified. Furthermore, there are still thousands of genes that have not been experimentally investigated in detail, leaving many genes where potential eye-related functions remain to be discovered. Indeed, systematic knockouts of 4,364 genes in mouse detected ocular phenotypes for 347 genes, with 75% of them not been known as eye-related before (*Moore et al., 2018*). This indicates that vision-related genes as well as genes associated with genetic eye disorders remain to be identified and characterized.

Interestingly, many genes that are linked to human eye diseases are inactivated (lost) in non-human mammals that naturally exhibit poor vision (*Emerling et al., 2017*; *Sharma and Hiller, 2020*). For example, subterranean mammals, such as the blind mole rat, naked mole rat, star-nosed mole, and cape golden mole, exhibit gene-inactivating mutations in genes implicated in cataract, retinitis pigmentosa, color or night blindness, or macular degeneration in human (e.g. *ABCA4*, *BEST1*, *CRYBA1*, *EYS*, *GJA8*, *GNAT2*, *PDE6C*, *ROM1*, and *SLC24A1*) (*Sharma and Hiller, 2020*; *Kim et al., 2011*; *Fang et al., 2014*; *Emerling and Springer, 2014*; *Prudent et al., 2016*; *Partha et al., 2017*; *Emerling, 2018*). Symptoms that characterize these human eye diseases resemble traits found in these subterranean mammals, such as highly degenerated retinae and lenses and sometimes blindness. Similarly, losses of the short wave sensitive opsin (*OPN1SW*), which is linked to color blindness in humans, occurred in several mammalian lineages such as cetaceans, bats, sloths, and armadillos that are consequently inferred to have monochromatic vision (*Emerling et al., 2017*; *Meredith et al., 2013*; *Zhao et al., 2009*; *Emerling and Springer, 2015*; *Sadier et al., 2018*; *Springer et al., 2016*). In addition to losses of vision-related genes, subterranean mammals also exhibit widespread sequence and transcription factor binding site divergence in eye-related regulatory elements (*Roscito et al., 2018*; *Langer et al., 2018*; *Langer and Hiller, 2019*). Such mutations can cause mis-expression of target genes in ocular tissues such as the lens (*Roscito et al., 2021*). Loss and divergence of vision-related genes and regulatory elements in these mammals is likely caused by the lack of natural selection on maintaining functional eyes in a dark environment. Taken together, previous studies established a clear association between naturally occurring poor vision phenotypes and regressive evolution of the genetic machinery required for functional eyes (*Emerling et al., 2017*; *Partha et al., 2017*; *Roscito et al., 2018*; *Langer et al., 2018*).

Here, we performed a genome-wide screen for genes preferentially lost in independent mammalian lineages with a low visual acuity with the goal of revealing currently uncharacterized genes, where vision-related gene loss patterns would predict vision-related functions. In addition to identifying losses of known vision-related genes, our screen revealed previously unknown losses of several functionally uncharacterized genes, among them *SERPINE3*, which is lost in 18 mammalian lineages that often do not use vision as the primary sense. We show that *SERPINE3* is specifically expressed in eyes of zebrafish and mouse. Furthermore, by knockout of *serpine3* in zebrafish, we show that the gene is required for maintenance of a proper retinal lamination and overall eye shape, which confirms the predicted vision-related function. Collectively, our results confirm that *SERPINE3* has functions in

vertebrate eyes and our discovery-driven study demonstrates how specific evolutionary divergence patterns can reveal novel insights into gene function (*Stephan et al., 2022*).

## Results

### A genome-wide screen retrieves genes preferentially lost in mammals with low visual acuity

To uncover potentially unknown vision-related genes, we used the Forward Genomics framework (*Hiller et al., 2012*) to search for associations between convergent gene losses and poor vision phenotypes in mammalian lineages, where poor vision has independently evolved. Vision is a complex multifaceted trait that may not be easily captured with a single variable. In our study, we decided to select mammals with poor vision based on visual acuity values for two reasons. First, visual acuity describes the ability of an animal to resolve static spatial details, which generally reflects how much an animal relies on vision in comparison to other senses (*Caves et al., 2018*). Second, visual acuity data is available for 49 placental mammals with sequenced genomes, enabling a comprehensive genomic screen (*Figure 1—source data 1*). Using visual acuity values, we defined two groups. Low-acuity species have low visual acuity values <1 ($\log_{10}$(va) <0), which comprises ten species (three echolocating bats, three rodents and four subterranean mammals) representing seven lineages (*Figure 1A*). All other species have visual acuity values >1 and comprise the group with higher visual acuity values.

Using this classification, we performed a genome-wide screen for genes that exhibit inactivating mutations (frame-shifting insertions or deletions, premature stop codons, splice site mutations, exon, or gene deletions) preferentially in low-acuity species, similar to previous screens (*Hecker et al., 2019a*; *Hecker et al., 2019b*; *Sharma et al., 2018*). Using a false discovery rate (FDR) cutoff of 0.05 and filtering for genes lost in at least three low-acuity lineages, we obtained 29 genes that are each convergently lost in at least three lineages of low-acuity mammals (*Figure 1B*, *Figure 1—source data 2*). These genes include several known vision-related genes, such as components of the photoreceptor signal transduction cascade (*GUCA1B*, *ARR3*), a factor required for retinal organization (*CRB1*), lens crystallins (*CRYBA1*, *CRYBB3*, *CRYGS*) and the cornea specific keratin 12 (*KRT12*). The set of 29 genes is enriched in vision-related functions such as the Gene Ontology term 'visual perception' (p=1.8e-7), expression in the mouse lens (p=5.5e-3), and human eye diseases (cataract: p=8.5e-3) (*Figure 1C*, *Figure 1—source data 3*). To confirm the specificity of these results, we performed a control screen for genes that are preferentially lost in high-acuity sister species of the low-acuity mammals. This control screen retrieved only two genes, none of which have known functions in the eye (*Figure 1—source data 4*). Together, this shows that our genome-wide screen for genes preferentially lost in low-acuity species successfully retrieved known vision-related genes.

Interestingly, while 15 of the 29 top-ranked hits have no studied role in the eye (*Figure 1B*), many of these genes are expressed in human eyes. *RSKR*, *LACTBL1*, *LBHD1*, and *ZNF529* even cluster in their expression pattern with other retina phototransduction and visual perception genes (*Uhlen et al., 2010*; *Figure 1—source data 2*). The preferential loss of these genes in species that exhibit lower visual acuity values and that have convergently lost other known vision-related genes predicts an uncharacterized vision-related function for some of these 15 genes (*Figure 1A*).

### *SERPINE3* is convergently lost in low-acuity mammals

We sought to experimentally test this prediction for a gene that is ranked highly in our screen. Since the first-ranked gene, *TSACC* (TSSK6 activating cochaperone), encodes a chaperone that is specifically expressed in testis (*Uhlen et al., 2010*; *Jha et al., 2010*) and is therefore unlikely to have an eye-related function, we focused on the second-ranked candidate, *SERPINE3* (serpin family E member 3). *SERPINE3* is largely uncharacterized and classified as lost in 7 of the 10 low-acuity mammals (*Figure 1*, *Figure 1—source data 2*). *SERPINE3* independently accumulated inactivating mutations in all four subterranean species (cape golden mole, star-nosed mole, naked mole rat, blind mole rat). Within bats, *SERPINE3* is inactivated in the two *Myotis* bats and the big brown bat (Yangochiroptera). These three species use echolocation instead of vision as the primary sense for hunting. In contrast, non-echolocating flying foxes (Pteropodidae) that rely more on vision, possess an intact *SERPINE3* gene. To show that *SERPINE3* loss is robustly associated with poor vision, we varied the visual acuity thresholds used to classify species as having low-acuity vision and explored an alternative approach for

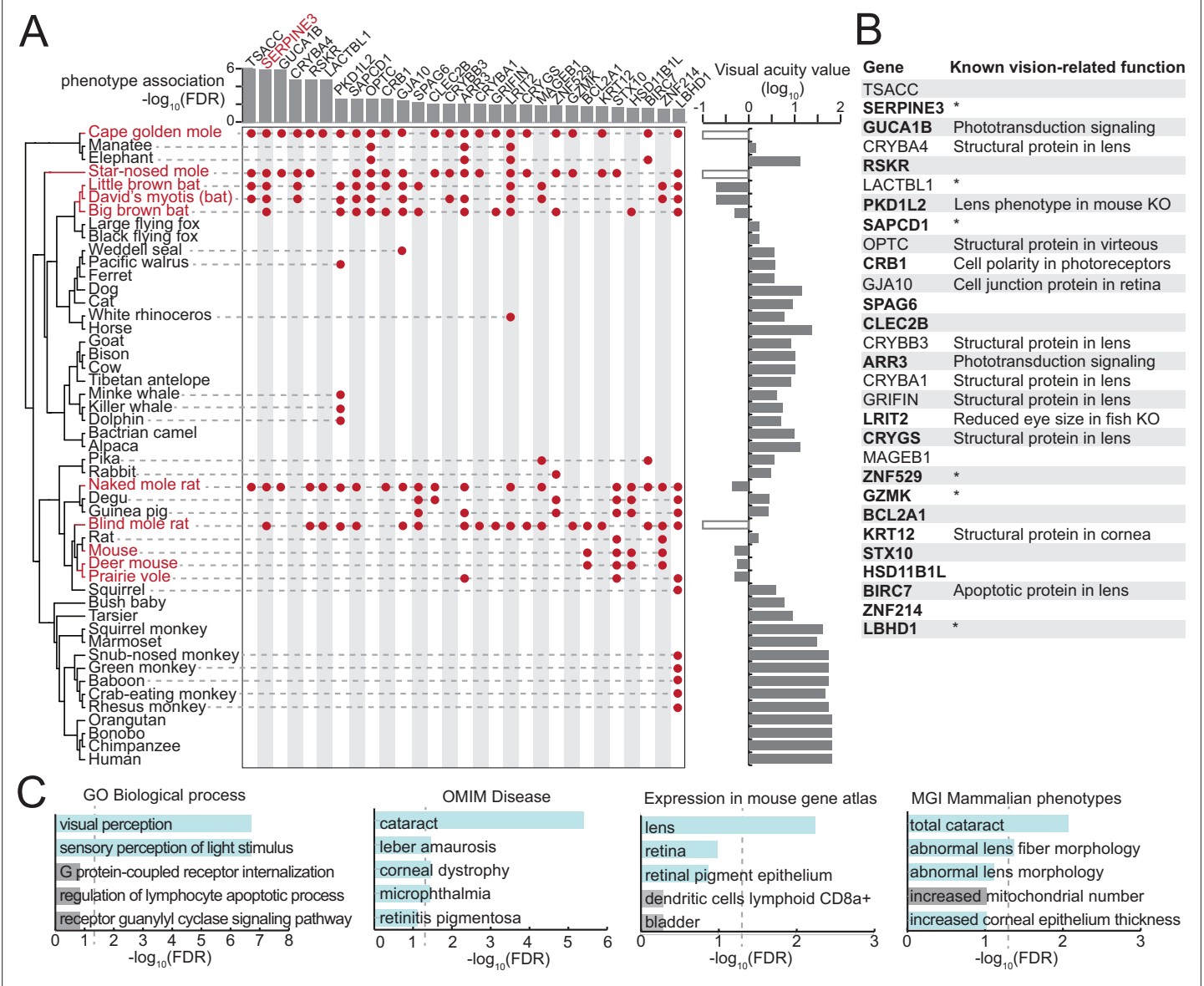

**Figure 1.** A comparative genomics screen uncovered known and novel vision-related genes. (**A**) Phylogeny of the species included in our screen (left). Visual acuity values on a $\log_{10}$ scale are shown on the right as a bar chart; white boxes indicate the three subterranean mammals that lack acuity measurements but are functionally blind. Low-acuity species (red font) are defined here as species with visual acuity <1 ($\log_{10}$(va) <0). At a false discovery rate (FDR) threshold of 0.05, our screen retrieved 29 genes, which are preferentially lost in low-acuity species. Gene losses in individual species are denoted by red dots. The FDR value for the gene loss – phenotype association is shown at the top as a bar chart. (**B**) List of 29 genes, together with known vision-related functions. Asterisk marks genes that have no known vision-related function but were mentioned in large-scale gene expression data sets of ocular tissues. Genes in bold are expressed in human eyes according to the eyeIntegration database (*Bryan et al., 2018*) (see Methods). KO - knockout. (**C**) Functional enrichments of the 29 genes reveals vision-related functions (Gene Ontology, GO of biological processes), associations with human eye disorders (OMIM), expression in ocular tissues in the mouse gene atlas (*Kuleshov et al., 2016*) and eye phenotypes in mouse gene KOs (MGI Mammalian Phenotype level 4) among the top five most significant terms. Vision-related terms are shown as blue bars. The dashed line indicates statistical significance in a one-sided Fisher's exact test after correcting for multiple testing using the Benjamini-Hochberg procedure (FDR 0.05).

The online version of this article includes the following source data and figure supplement(s) for figure 1:

**Source data 1.** Genome assemblies used in this study and visual acuity values of mammalian species.

**Source data 2.** List of candidate genes of the forward genomics screen based on visual acuity <1 including a short description.

**Source data 3.** Enrichment of candidate genes in different processes/functions.

**Source data 4.** A control forward genomics screen does not retrieve vision-related genes.

*Figure 1 continued on next page*

*Figure 1 continued*

**Source data 5.** List of candidate genes of the forward genomics screen based on a visual acuity threshold <1.5.

**Source data 6.** List of candidate genes of the forward genomics screen based on visual acuity threshold <0.5.

**Source data 7.** List of candidate genes of the forward genomics screen based on the molecular loss signature of known vision genes.

**Source data 8.** Reading frame %intact values for all investigated genes and species.

**Source data 9.** Enrichment of candidate genes in different stages of the filtering process.

**Figure supplement 1.** Flow diagram of filtering steps applied in the Forward genomics approach.

---

defining poor vision species based on a molecular loss signature of known vision-related genes. All three modified screens consistently retrieved *SERPINE3* as one of the top-ranked hits (*Figure 1—source data 5* , *Figure 1—source data 6*, *Figure 1—source data 7*), showing that this association is robust to the selected thresholds and phenotype definition.

## *SERPINE3* became dispensable in many mammals that do not primarily rely on vision

To explore the evolution of *SERPINE3* in additional mammalian genomes, we made use of an orthology data set generated by the TOGA (Tool to infer Orthologs from Genome Alignments) method (Kirilenko et al., titled 'Integrating gene annotation with orthology inference at scale', under review) that includes 381 additional placental mammalian species that were not part of the genomic screen. Interestingly, this substantially extended data set revealed a number of additional losses of *SERPINE3*, typically in species that do not rely on vision as their primary sense (*Figure 2A*). We only considered *SERPINE3* losses for which at least one inactivating mutation could be validated (see Materials and methods, *Figure 2—source data 1*, *Figure 2—source data 2*). Further supporting *SERPINE3* losses, analyses of publicly available RNA-seq data indicates that remnants of inactivated *SERPINE3* genes do not show relevant expression (*Figure 2—figure supplement 1*, *Figure 2—source data 3*). A detailed analysis of inactivating mutations indicates that *SERPINE3* is inactivated in 70 of the 430 analyzed placental mammals and that the gene is convergently lost at least 18 times in placental mammal evolution (*Figure 2—figure supplements 2–8*).

For example, *SERPINE3* is lost in several mammalian clades with nocturnal representatives that have a partially burrowing lifestyle such as pangolins (Manidae), aardvark and armadillos (Cingulata), which are characterized by proportionally small eyes (*Mittermeier and Wilson, 2011*; *Myers et al., 2022*; *Figure 2A*, *Figure 2—figure supplements 2 and 3*). All representatives of these clades use smell and hearing as primary senses for perception of environmental clues and have a poor sense of vision (*DiPaola et al., 2020*). *SERPINE3* has also been lost in the stem lineage of Pilosa (sloths and anteaters). Although extant sloths have rather high visual acuity values (10.2 for southern two-toed sloth) (*Veilleux and Kirk, 2014*), the Pilosa ancestor was likely a digging/burrowing species (*Gaudin and Croft, 2015*), indicating that *SERPINE3* loss originally occurred in a burrowing species. *SERPINE3* is also lost in solenodon, European shrew and tenrecs (Tenrecidae), species with small eyes that use echolocation for close-range spatial orientation (*Siemers et al., 2009*; *Gould, 1965*; *Eisenberg and Gould, 1966*; *Figure 2B*, *Figure 2—figure supplements 2 and 4*). For bats, our extended data set shows that *SERPINE3* is convergently lost in seven echolocating lineages, but remains intact in all 13 analyzed Pteropodid bats, revealing a clear pattern of convergent losses of this gene restricted to bat lineages relying on laryngeal echolocation (*Figure 2—figure supplements 5–7*). The sac-winged bat is the only laryngeal echolocating bat with an intact *SERPINE3*; however, selection rate analysis indicates that the gene evolves under relaxed selection (*Figure 2—source data 4*). The available genomes support shared gene-inactivating mutations among many bat species, which likely represent ancestral events (*Figure 2A* right). Expanding upon the two convergent losses of *SERPINE3* in subterranean rodents detected in the initial screen, the gene is also convergently lost in the fossorial Damara mole rat and evolves under relaxed selection in the Transcaucasian mole vole (*Figure 2—source data 4*, *Figure 2—figure supplement 8*), while being intact in all other 63 rodents. In the expanded data set, we further uncovered the loss of *SERPINE3* in another clade of fossorial moles (Talpidae) within the order of Eulipotyphla (*Figure 2—figure supplement 4*). A splice site mutation shared between dugong and manatee as well as patterns of relaxed selection indicate an ancestral loss of *SERPINE3* in the ancestor of sirenians, which mostly use their tactile sense for navigation through murky water

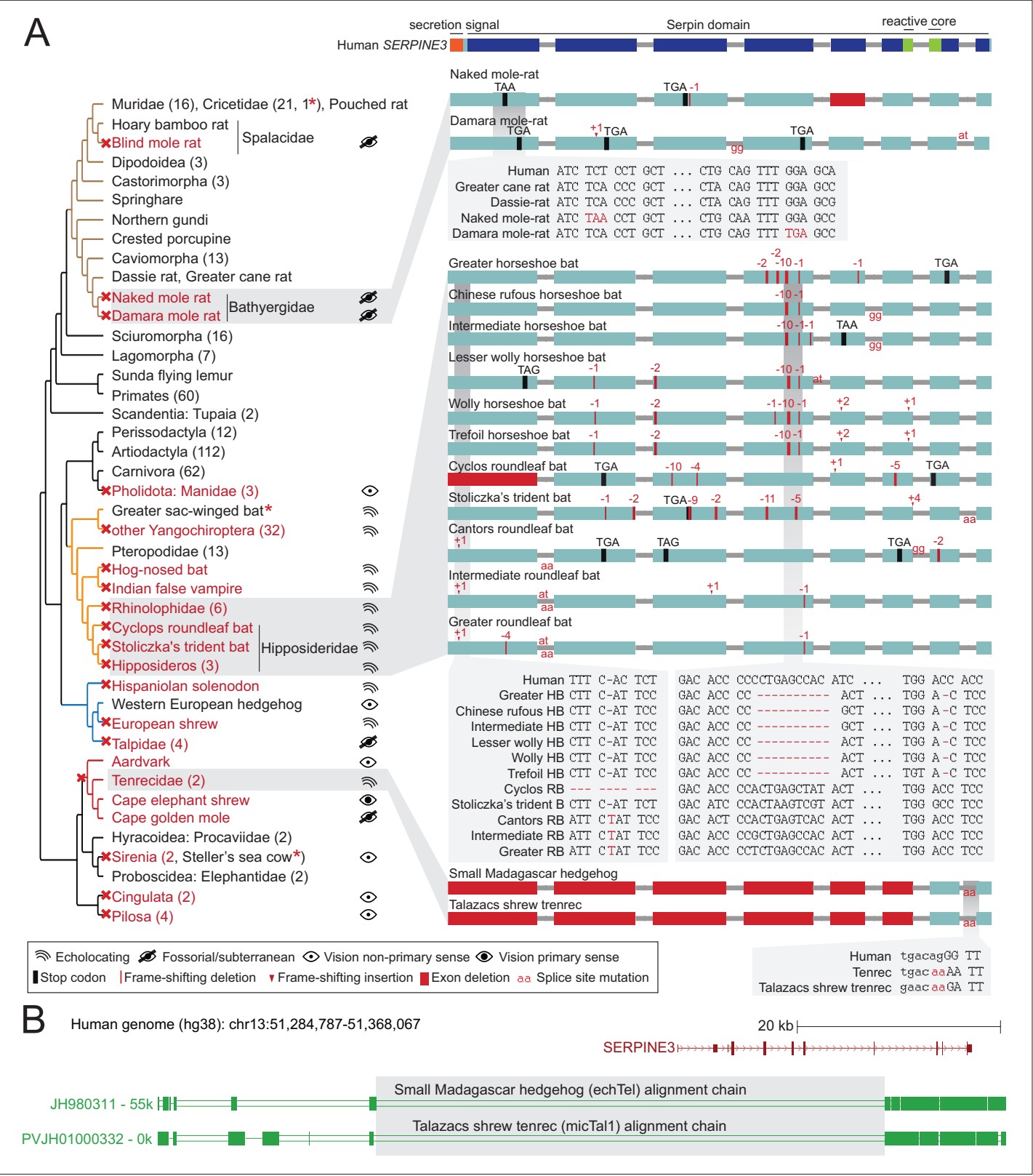

**Figure 2.** *SERPINE3* gene loss pattern across 430 mammalian species. (**A**) Left: Phylogeny of mammalian species investigated for the loss of *SERPINE3* with mapped gene loss events indicated as red crosses. Branches of major clades are colored (Rodentia – brown, Chiroptera – orange, Eulipotyphla – blue, Afroinsectiphilia - red). The number of species investigated per clade is specified in parenthesis. For all loss lineages (red font), visual capability (classified as echolocating, fossorial/subterranean, vision as non-primary and primary sense) is displayed as pictograms at the right. Asterisk marks

*Figure 2 continued*

indicate species, where *SERPINE3* evolved under relaxed selection but did not accumulate not more than one inactivating mutation. Right: The Serpin protein domain (Pfam) spans all eight protein-coding exons (boxes) of the intact human *SERPINE3* gene (top). The secretion signal and the reactive core region are conserved in species with an intact *SERPINE3* (*Figure 2—figure supplements 9 and 10*). Gene-inactivating mutations are illustrated for three clades, with stop codon mutations shown in black, frame-shifting insertions and deletions shown in red and mutated splice site dinucleotides shown between exons in red. Deleted exons are shown as red boxes. Insets show codon alignments with inactivating mutations in red font. RB - roundleaf bat, HB - horseshoe bat. (**B**) An ancestral deletion removed large parts of *SERPINE3* in the tenrec lineage. UCSC genome browser (***Lee et al., 2022***) view of the human hg38 assembly showing the *SERPINE3* locus and whole genome alignments between human and two tenrec species, visualized as a chain of co-linear local alignments. In these chains, blocks represent aligning sequence and double lines represent sequence between the aligning blocks that do not align between human and tenrec. A large deletion removed the first five protein-coding exons of *SERPINE3* in both species. Shared breakpoints (gray box) indicate that the deletion likely represents an ancestral event in Tenrecidae.

The online version of this article includes the following source data and figure supplement(s) for figure 2:

**Source data 1.** *SERPINE3* intactness in 447 mammalian assemblies.

**Source data 2.** Verification of inactivating mutations in *SERPINE3* with raw DNA sequencing reads.

**Source data 3.** Expression of *SERPINE3* remnants in public RNA-seq datasets.

**Source data 4.** Analysis of relaxation or intensification of selection in *SERPINE3* branches of interest.

**Figure supplement 1.** The remnants of *SERPINE3* are not expressed in the common vampire bat.

**Figure supplement 2.** Gene-inactivating mutations support four independent losses of *SERPINE3* in Afrotheria and Xenarthra (red dots).

**Figure supplement 3.** Gene-inactivating mutations support a single loss of *SERPINE3* in Pholidota.

**Figure supplement 4.** Gene-inactivating mutations support three independent losses of *SERPINE3* in Eulipotyphla (red dots).

**Figure supplement 5.** Gene-inactivating mutations support one loss of *SERPINE3* in Yangochiroptera.

**Figure supplement 6.** Gene-inactivating mutations in additional Yangochiroptera.

**Figure supplement 7.** Gene-inactivating mutations support six independent losses of *SERPINE3* in Yinpterochiroptera.

**Figure supplement 8.** Independent gene-inactivating mutations in other mammals.

**Figure supplement 9.** Conservation of the signal peptide in mammalian SERPINE3.

**Figure supplement 10.** The hinge region and reactive core loop are conserved in intact mammalian SERPINE3.

---

(***Moore et al., 2021***; *Figure 2—figure supplement 2*, *Figure 2—source data 1*). Consistent with our definition of poor vision based on visual acuity, the Florida manatee has a reduced ability to resolve spatial detail (va = 1.6, ***Mass et al., 2012***) compared to its closest relatives, the African elephant (va = 13.16, ***Pettigrew et al., 2010***), which has an intact *SERPINE3* gene. Finally, *SERPINE3* is also lost in the cape elephant shrew, which is a mostly diurnal species with relatively large eyes (***Dengler-Crish et al., 2006***); however, loss of *SERPINE3* already occurred in the ancestral Afroinsectiphilia lineage (*Figure 2—figure supplement 2*), which was presumably nocturnal (***Wu et al., 2017***).

Together, many independent gene losses in species that do not rely on vision as their primary sense predicts a vision-related function for *SERPINE3* that became dispensable in these mammals, leading to convergent *SERPINE3* losses due to relaxed selection.

## *SERPINE3* encodes a putative secreted proteinase inhibitor

Based on sequence homology, *SERPINE3* is classified as a member of the **ser**ine **p**roteinase **in**hibitor (SERPIN) family. Many members of this family are secreted into the extracellular space and inhibit their substrates by covalent binding (***Heit et al., 2013***; ***Law et al., 2006***). We performed a SERPINE3 sequence analysis, which revealed that key sequence features of inhibitory serpins are well conserved among placental mammals, suggesting that SERPINE3 also functions as a secreted serine proteinase inhibitor (see Material and Methods, *Figure 2—figure supplements 9 and 10*).

Serpins have roles in coagulation, angiogenesis, neuroprotection and inflammation, and several serpins have been implicated in human diseases (***Law et al., 2006***; ***Barnstable and Tombran-Tink, 2004***). However, the functional role of *SERPINE3* is largely unknown as the gene has never been studied in an animal or cellular model. A thorough literature search revealed that *SERPINE3* is listed (often in the supplement) among many other genes, as differentially expressed in large-scale expression analyses. For example, *Serpine3* is upregulated in the mouse retina in response to overexpressing neuroprotective factors (***Machalińska et al., 2013***). *Serpine3* was also upregulated in complement component 3 (*C3*) knockout mice, which represent a model of the aged retina (***Rogińska et al., 2017***),

and downregulated in the eye of knockout mice for *Pcare*, a causal gene for retinitis pigmentosa (*Kevany et al., 2015*). Together, while this gene was never studied in greater detail, literature clues and the striking convergent gene loss pattern in mammals with poor vision suggest that *SERPINE3* may have a functional role in the eye.

## *Serpine3* is specifically expressed in Müller glia in the adult zebrafish retina

To test the prediction that *SERPINE3* has an eye-related function, we first analyzed its expression pattern in adult zebrafish (*Figure 3*). Zebrafish has proven as valuable model species for the study of eye genetics as it has a cone-dominated retina (~60% cones, ~40% rods) (*Fadool, 2003*), similar to the central human retina. Furthermore, zebrafish have a single *serpine3* ortholog that is located in a context of conserved gene order (*Figure 4A*), which makes zebrafish a suitable model for investigation of *serpine3* expression and function. Reverse transcription quantitative PCR (RT-qPCR) analysis with two normalization genes (*actb* and *rpl13a* with n=8 and n=3 biological replicates, respectively) consistently revealed that the highest *serpine3* expression is in the eye, followed by expression in the brain (*Figure 3A and B*, *Figure 3—source data 1*, *Figure 3—source data 2*). For *actb*, *serpine3* expression is significantly higher in eye than in other tissues (*Figure 3A*).

To better characterize *serpine3* expression in the eye, we next performed RT-qPCR on samples of RPE, retina and eye without retina and RPE (*Figure 3C*, *Figure 3—source data 3*). *Serpine3* is expressed significantly higher in the retina compared to RPE only (two-sided unequal variances t-test, p=0.01, *Figure 3C*). Expression is barely detectable in whole eye without retina and RPE (two-sided unequal variances t-test, eye vs. retina, p=0.009, *Figure 3C*), suggesting that the expression signal obtained from whole eye RT-qPCR mostly originates from expression in retina.

To specify in which cell types *serpine3* is expressed, we finally performed in situ hybridization (ISH) on adult zebrafish retinae. *Serpine3* expression is detected in the inner nuclear layer throughout the retina with stronger signals in the ventral region close to the optic nerve (*Figure 3E and F*). Co-staining with different cell type specific markers reveals expression of *serpine3* in a fraction of glial fibrillary acidic protein (*gfap*)-positive Müller glia (MG) cells (*Figure 3G and H*, *Figure 3—figure supplement 1*), whereas expression of *serpine3* mRNA was not detected in bipolar or amacrine cells (*Figure 3—figure supplement 2*).

To also explore *SERPINE3* expression in mammals, we first used RT-qPCR to analyze *Serpine3* expression in mouse. Similar to zebrafish, mouse *Serpine3* expression is highest in whole eye, whereas expression was not detectable in colon, cortex, heart, liver, spleen, and testis (*Figure 3D*, *Figure 3—source data 4*). Next, we analyzed *SERPINE3* expression in other mammals and vertebrates using publicly available expression data sets. Despite sparser data, we found evidence for expression of *SERPINE3* in eyes of human, rat, cat, cow and chicken, which suggests a conserved expression pattern and a role in vertebrate vision (*Figure 3—source data 5*).

In summary, our experiments in zebrafish and mouse together with available data sets of different vertebrates show that *SERPINE3* is expressed in the vertebrate eye, specifically in zebrafish in MG cells.

## Knockout of *serpine3* in zebrafish leads to morphological defects in the eye including the retina

Next, we tested whether *serpine3* inactivation results in an eye phenotype. To this end, we deleted the transcription start site of *serpine3* with CRISPR-Cas9, denoted as the *serpine3*cbg17 allele (*Figure 4A*, *Figure 4—figure supplements 1 and 2*). Adult homozygous *serpine3*cbg17 fish were viable and fertile. Using RT-qPCR and ISH, we confirmed that the deletion of the transcription start site abolished *serpine3* expression in retinae of adult homozygotes for *serpine3*cbg17 compared to wild type (WT) siblings (two-sided unequal variances t-test, p=0.049, *Figure 4B and C*, *Figure 4—source data 1*).

We found that adult *serpine3*cbg17 individuals frequently showed notches in the iris of one of the eyes (4 of 5 individuals, white arrows in *Figure 4D*, *Figure 4—source data 2*, source data on Dryad), which affected the eye's overall shape. To confirm that these notch-caused shape deviations are caused by inactivation of *serpine3*, we generated an independent line, where we introduced an early frameshift in *serpine3's* coding exon 1 with CRISPR-Cas9, denoted as the *serpine3*cbg18 allele (*Figure 4A*, *Figure 4—figure supplements 1 and 2*). *Serpine3*cbg18 fish also showed a high frequency of notches

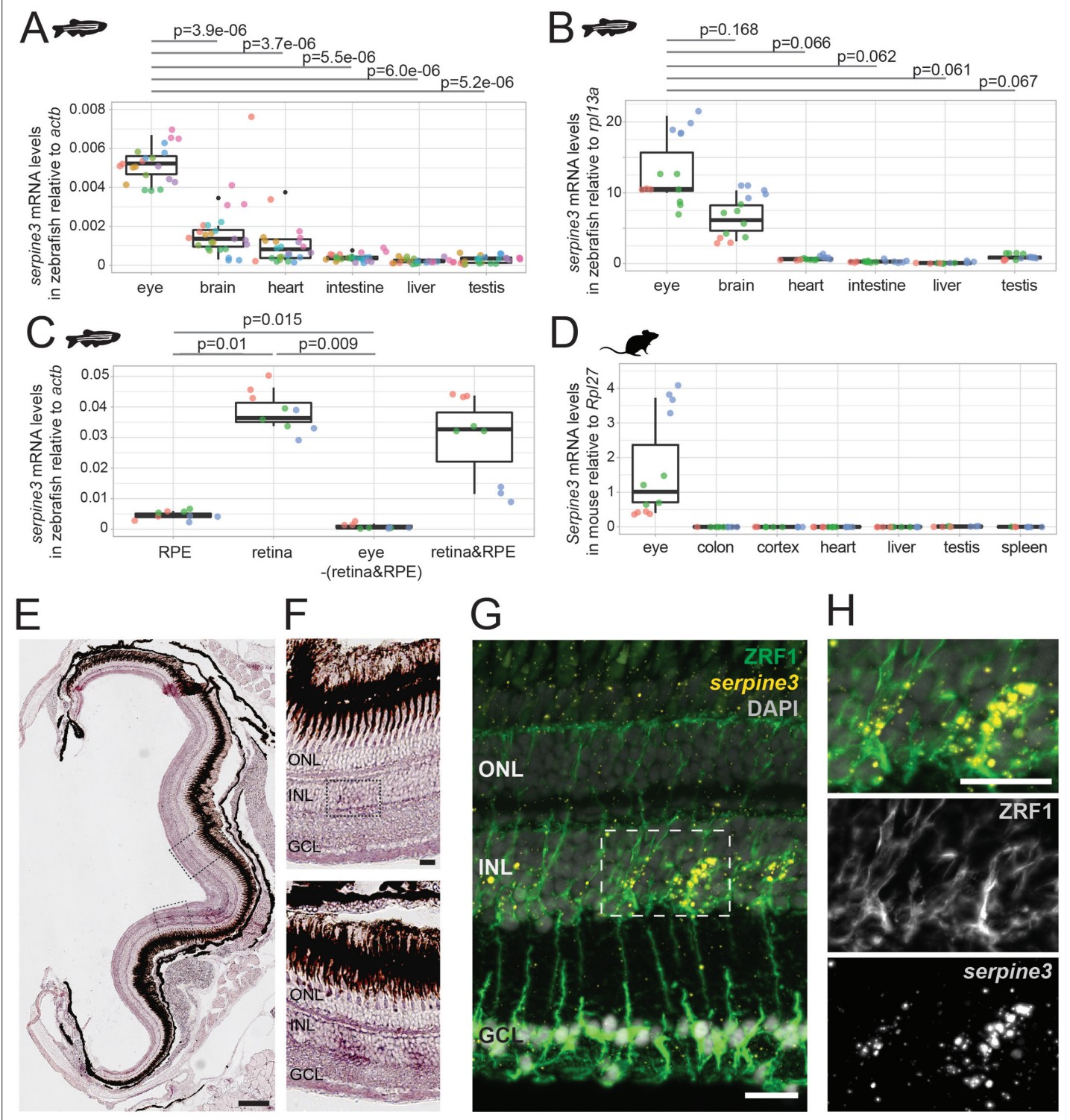

**Figure 3.** *Serpine3* is expressed in zebrafish and mouse eyes. (**A,B**) Expression of zebrafish *serpine3* mRNA in relation to the normalization genes *actb* (**A**) and *rpl13a* (**B**) measured with RT-qPCR. Expression levels are consistently the highest in eye. A two-sided unequal variances t-test was used. Boxplots display first quartile, median and third quartile with whiskers extending to the maximum and minimum value. Measurements of technical replicates (n=3–6) are encoded in the same color, different colors represent different biological replicates (n=3–8). Relative rates are displayed as zero, if expression was below the detection limit. (**C**) *Serpine3* mRNA expression in different tissues of the zebrafish eye in relation to the reference gene *actb* measured with RT-qPCR for three biological replicates. *Serpine3* is specifically expressed in the retina but not in other tissues of the eye. The expression level is significantly higher in retina without RPE compared to RPE only (two-sided unequal variances t-test). Although expression in RPE only is low, it is

*Figure 3 continued on next page*

*Figure 3 continued*

significantly higher than expression in eye tissue after removing retina and RPE (two-sided unequal variances t-test). (**D**) *Serpine3* mRNA expression in mouse in relation to the reference gene *Rpl27* measured with RT-qPCR for three biological replicates. *Serpine3* is specifically expressed in the eye but not in colon, cortex, heart, liver, spleen and testis. (**E–H**) *Serpine3* mRNA expression pattern in zebrafish retina. (**E–F**) Chromogenic in situ hybridization (ISH) shows localized expression of *serpine3* (purple) in the retina, specifically in the inner nuclear layer (inlet). (**G–H**) Fluorescence in situ hybridization shows that *serpine3* mRNA expression (yellow) is localized to cell bodies of Müller glia cells. Filaments of Müller glia cells are marked by the glial fibrillary acidic protein (ZRF1 antibody, green). Specificity of the *serpine3* ISH probe is shown by absence of the signal in homozygous *serpine3*cbg17 knockout fish (*Figure 3—figure supplement 1*). *Serpine3* mRNA is not expressed in bipolar or amacrine cells (*Figure 3—figure supplement 2*). Nuclei are stained with DAPI (white). Scale bar is 200 µm in (**E**) and (**G**) and 20 µm in (**F**) and (**H**). INL – inner nuclear layer, ONL – outer nuclear layer, GCL – ganglion cell layer.

The online version of this article includes the following source data and figure supplement(s) for figure 3:

**Source data 1.** qPCR analysis of a zebrafish organ series using the normalizer gene *actb*.

**Source data 2.** qPCR analysis of a zebrafish organ series using the normalizer gene *rpl13a*.

**Source data 3.** qPCR analysis of zebrafish eye subtissues.

**Source data 4.** qPCR analysis of mouse organs.

**Source data 5.** Data sets supporting expression of *SERPINE3* in tissues of the eye in vertebrates other than mouse and zebrafish.

**Figure supplement 1.** *Serpine3* expression co-localizes with glial fibrillary acidic protein in the retina of wild type fish.

**Figure supplement 2.** *Serpine3* expression does not co-localize with markers for bipolar or amacrine cells in the zebrafish retina.

in 13 of 20 individuals (*Figure 4—source data 2*, source data on Dryad). To quantify this shape deviation, we calculated iris solidity, which compares the area ratio of the eye's outer shape (white line) and its concave shape (red dotted line in *Figure 4D*). Eyes of both *serpine3*cbg17 and *serpine3*cbg18 individuals had a significantly reduced solidity in comparison to their WT siblings (Wilcoxon rank sum test, p=0.009 and p=6.35e-9, respectively, *Figure 4E*). Iris circularity, another descriptor of the eye shape deviation, is also significantly reduced in homozygous individuals of both lines (*Figure 4—figure supplement 3*). Together this indicates that mutations in *serpine3* cause eye shape deviations in zebrafish.

We next investigated the ocular morphology of adult fish of both lines and their WT siblings using hematoxylin/eosin sections (*Figure 4F*; *Figure 4—figure supplements 4 and 5*). While ocular structure and optic nerves of all inspected eyes are largely normal (*Figure 4—figure supplement 4*), the distance between retina and lens is reduced in homozygous fish of both alleles (*Figure 4F*, *Figure 4—figure supplement 5*). A detailed inspection of the retinal organization revealed that although all retinal layers were present, the mutant retinae are generally less organized and structured compared to WT siblings. Retinal cells appear reduced in number and less densely packed. Most prominently, we noticed that rod outer segments and the pigmented RPE cells, which were aligned in WT fish, were not clearly separated in homozygous fish of both alleles (*Figure 4Fc-e*, *Figure 4—figure supplement 5*). Furthermore, we observed large clusters of pigmented cells in the photoreceptor layer (empty arrows, *Figure 4Fc-e*) as well as single displaced pigmented cells in all retinal layers (yellow arrows, *Figure 4Fc-e*). Similar alterations in retinal structure were detected in homozygous *serpine3*cbg18 fish (*Figure 4—figure supplement 5*). This shows that KO of *serpine3* in zebrafish results in morphological defects in the eye, characterized by differences in eye shape and retinal organization.

## Polymorphisms near human *SERPINE3* are associated with human eye phenotypes

We next analyzed recently published Genome Wide Association Study data for human single nucleotide polymorphisms (SNPs) associated with eye-related traits. This analysis revealed two such SNPs that are in linkage disequilibrium with *SERPINE3* (*Figure 4A*). rs1028727 is located ~10 kb upstream of the *SERPINE3* transcription start site and is associated with a decreased area of the optic nerve head (*Bonnemaijer et al., 2019*). rs7327381 is located ~97 kb downstream of the *SERPINE3* transcription start site and is associated with an increase in corneal curvature (*Fan et al., 2020*). This suggests that human SNPs linked to *SERPINE3* are associated with eye phenotypes, supporting a putative ocular function of human *SERPINE3*.

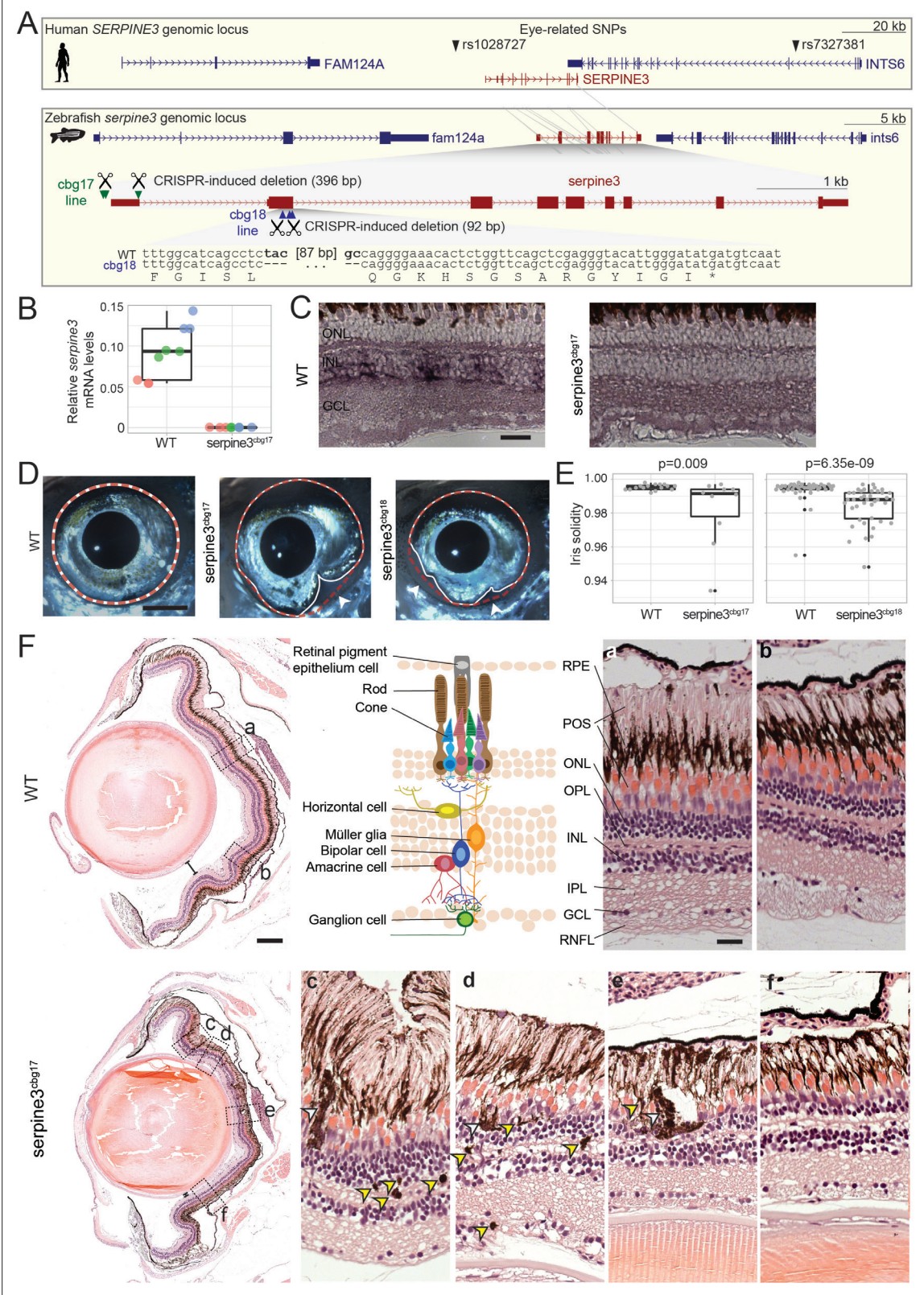

**Figure 4.** *Serpine3* knockout in zebrafish causes defects in eye shape and retinal layering. (**A**) UCSC genome browser visualization of the *SERPINE3* genomic locus in human (hg38 assembly, top box) and zebrafish (danRer11 assembly, bottom box) shows that both species have a 1:1 ortholog with the same number of coding exons in a conserved gene order context. In the human locus, two single nucleotide polymorphisms (SNPs) are in linkage with *SERPINE3* and associated with eye phenotypes. In zebrafish, we used CRISPR-Cas9 to generate two independent knockout (KO) lines. The position of

*Figure 4 continued on next page*

*Figure 4 continued*

guide RNAs is indicated as scissors. In the *serpine3*<sup>cbg17</sup> line, we deleted the promoter and first exon. In the *serpine3*<sup>cbg18</sup> line, we introduced a 92 bp frame-shifting deletion in exon 2 (coding exon 1) that results in three early stop codons in the original reading frame. (**B**) Relative expression of *serpine3* mRNA in wild type (WT) zebrafish and *serpine3*<sup>cbg17</sup> individuals quantified by RT-qPCR relative to the expression of *rpl13a*. *Serpine3* mRNA expression is close to zero in *serpine3*<sup>cbg17</sup> fish and significantly reduced in comparison to wild type fish (p=0.049, two-sided unequal variances t-test). Technical replicates of the qPCR are shown as individual data points; different colors represent different biological replicates. Boxplots display first quartile, median and third quartile with whiskers extending to the maximum and minimum of the three biological replicates. (**C**) In situ hybridization showing that *serpine3* is expressed in the inner nuclear layer (INL) of WT zebrafish but not in the homozygous *serpine3*<sup>cbg17</sup>. Scale bar = 25 μm. (**D**) *Serpine3* knockout leads to changes in eye shape in adult, homozygous knockout (KO) fish of *serpine3*<sup>cbg17</sup> and *serpine3*<sup>cbg18</sup> lines in comparison to their WT siblings (18 and 11 months, respectively). In WT, the eye shape almost perfectly corresponds to the concave shape of the iris (overlay of white and red dotted lines). In contrast, many KO individuals have alterations in eye shape, evident by notches (arrow heads) in the white line that follows the iris. Scale bar = 1 mm. (**E**) Iris solidity (ratio of eye shape/ concave eye shape) significantly differs between WT and KO siblings for both the *serpine3*<sup>cbg17</sup> (16 vs 10 eyes) and the *serpine3*<sup>cbg18</sup> (40 vs 40 eyes) line. A Wilcoxon Rank sum test was used. Boxplots display first quartile, median, and third quartile with whiskers extending to the maximum and minimum within 1.5 times interquartile range. Outliers are shown in black. Iris circularity, another quantification measure for the phenotype, is shown in *Figure 4—figure supplement 3*. (**F**) Hematoxylin/eosin histology staining of the eye of *serpine3*<sup>cbg17</sup> fish (22 months) reveals histological differences in comparison to their WT siblings (dorsal top, ventral bottom). In comparison to WT, distance between lens and retina of *serpine3*<sup>cbg17</sup> fish is reduced (distance bars). The WT retina (top) has a distinct lamination with clear separation of the single retinal layers (**a, b**) as shown in the schematic (RPE – retinal pigment epithelium layer, POS – photoreceptor outer segment, ONL – outer nuclear layer, OPL – outer plexiform layer, INL – inner nuclear layer, IPL – inner plexiform layer, GCL – ganglion cell layer, RNFL – retinal nerve fiber layer). Although all retinal layers are present in *serpine3*<sup>cbg17</sup> fish, the layering appears distorted and the density of cells is reduced (**c–f**). Specifically, the RPE cells display an altered distribution and even local clusters (empty arrows), and displaced pigmented cells emerge in all retinal layers (yellow arrows). This was confirmed also for the cbg18 allele (*Figure 4—figure supplement 5*). Scale bar in the overviews = 200 μm, scale bar in the magnifications = 20 μm.

The online version of this article includes the following source data and figure supplement(s) for figure 4:

**Source data 1.** qPCR analysis of zebrafish *serpine3*<sup>cbg17</sup> homozygotes (KO) vs WT.

**Source data 2.** Macroscopic eye phenotype in *serpine3* knockout lines.

**Source data 3.** Single-nucleotide polymorphism near SERPINE3 (13: 51, 341, 032–51, 362,101) are associated with human eye phenotypes.

**Source data 4.** Raw gel images of genotyping of the *serpine3*<sup>cbg17</sup> line with two primers (labeled image is shown in Figure 4 - figure supplement 2A).

**Source data 5.** Raw gel images of genotyping of the *serpine3*<sup>cbg17</sup> line with a mix of three primers (labeled image is shown in Figure 4 - figure supplement 2B).

**Source data 6.** Raw gel images of genotyping of the *serpine3*<sup>cbg18</sup> line with two primers (labeled image is shown in Figure 4 - figure supplement 2C).

**Figure supplement 1.** Knockout of *serpine3*<sup>cbg17</sup> and *serpine3*<sup>cbg18</sup> alleles in zebrafish by CRISPR-Cas9 is confirmed by sequencing results.

**Figure supplement 2.** PCRs confirm expected CRISPR-Cas9 induced deletions in *serpine3*<sup>cbg17</sup> and *serpine3*<sup>cbg18</sup> fish.

**Figure supplement 3.** Iris circularity differs between *serpine3*<sup>cbg17</sup> and *serpine3*<sup>cbg18</sup> fish and their respective wild type siblings (WT).

**Figure supplement 4.** The optic nerve is intact in *serpine3*<sup>cbg17</sup> and *serpine3*<sup>cbg18</sup> fish as shown with a hematoxylin/eosin histology staining.

**Figure supplement 5.** Hematoxylin/eosin histology staining of *serpine3*<sup>cbg18</sup> eyes (14 months) reveals histological differences in comparison to their wild type (WT) siblings (dorsal top, ventral bottom).

## Discussion

Our study combines comparative genomics to predict genes having vision-related functions and experiments in zebrafish to confirm this prediction for a top-ranked candidate, the uncharacterized *SERPINE3* gene. By conducting a genome-wide screen for genes that are preferentially lost in mammals with low visual acuity values, we uncovered both known vision-related genes as well as several genes that have no known eye-related function. One of the top-ranked candidates is *SERPINE3*, which we found to be independently lost at least 18 times in mammalian evolution, preferentially in species that do not use vision as the primary sense. For mouse and zebrafish, we show that the highest *SERPINE3* expression is in the eye, which is corroborated by available expression data of other vertebrates. By generating the zebrafish *serpine3*<sup>cbg17</sup> and *serpine3*<sup>cbg18</sup> knockout (KO) lines, we show that inactivation of this gene results in abnormal eye shape and retinal lamination, revealing an eye-related function for *serpine3*. This is further supported by *SERPINE3*-linked polymorphisms that are associated with eye-related traits in human.

In zebrafish, *serpine3* is expressed in a fraction of Müller glia (MG) cells. MG are the major retinal macroglia and perform numerous functions. By removing waste products and secreting (neuro) trophic substances and signaling molecules, they maintain the blood-retinal barrier and regulate

vascularization (*Reichenbach and Bringmann, 2013*). Most importantly, MGs are essential for the long-term viability of photoreceptors and other neuronal cell types (*Bringmann et al., 2009*). Our *serpine3* KO fish exhibit a disorganization of RPE cells, a phenotype that resembles those previously observed in experiments that perturb MG function. For example, selective ablation of MG in adult mice results in eye defects including aggregation of RPE cells and displacement of pigment granules in the ganglion cell layer (*Shen et al., 2012*). Interestingly, we also observed that *serpine3* KO affects eye shape, which cannot be readily explained by a direct MG-mediated effect. However, MG span the entire retina, connecting the extracellular space of retinal neurons, the vitreous and the capillaries at the apical retina (*Reichenbach and Bringmann, 2013*; *Klimczak et al., 2009*). It is therefore possible that a secreted protein, as predicted for serpine3 based on its conserved signal peptide, may also affect other eye subtissues. Finally, in zebrafish, MG are able to regenerate retinal neurons upon injury. However, *serpine3* does not seem to be involved in this process as during stress-induced regeneration, it is upregulated in resting MG that do not proliferate (*Hoang et al., 2020*).

In mammals, the *SERPINE3* gene loss pattern and expression profile in several species also support an eye-related function; however, the cell type expression pattern may differ between mammals and zebrafish. While our experiments in zebrafish show *serpine3* expression in MG, which is in agreement with *Hoang et al., 2020*, in mouse and human, *SERPINE3* seems to be expressed in RPE (*Hoang et al., 2020*; *Hadziahmetovic et al., 2012*; *Cowan et al., 2020*) indicating that cell type specificity may differ between mammals and fish. As *SERPINE3* is likely a secreted extracellular protein, it is possible that it has a similar function in mammals and zebrafish, despite secretion by different cell types. Whether this is the case or whether *SERPINE3* function and protein expression pattern in the mammalian eye differs remains to be explored in future studies. At the molecular level, this question may be addressed by investigating whether the molecular targets of *SERPINE3* are conserved among vertebrates.

Of particular interest is elucidating the functional role of *SERPINE3* in human. Several pieces of evidence indicate a potential role of this gene in anti-inflammatory processes and retinal survival. *SERPINE3* is upregulated in human patients with age-related macular degeneration (*Newman et al., 2012*), a progressive eye disease that is linked to chronic inflammation and wound healing. Consistent with a role in retinal survival, mouse *Serpine3* is upregulated after experimental overexpression of neurotrophin-4, a neuroprotective factor that promotes retinal survival (*Machalińska et al., 2013*). Furthermore, *SERPINE3* is a hallmark gene of differentiated, healthy human RPE cells (*Radeke et al., 2015*; *Boles et al., 2020*; *Sripathi et al., 2021*) that also have neurotrophic functions in the retina. It is thus conceivable that perturbation of proper *SERPINE3* expression or function may influence age-related diseases or human eye phenotypes, as indicated by human polymorphisms that are linked to *SERPINE3*.

In addition to *SERPINE3*, our screen also revealed other candidate genes for which unknown eye-related functions are plausible. The *LACTBL1* gene encodes a putative serine proteinase (*Liobikas et al., 2006*) and has an expression pattern similar to retina phototransduction genes (*Uhlen et al., 2010*). Furthermore, 20 kb upstream of *LACTBL1* is a linked SNP (rs10158878) that is associated with refractive error in human (GWAS catalog, *Hysi et al., 2020*). Another uncharacterized candidate gene with a strong visual acuity associated loss pattern is *SAPCD1*. A missense mutation (rs6905572) within *SAPCD1* is associated with macular degeneration (dbSNP, *Chen et al., 2010*). The *LRIT2* gene is linked to SNPs (rs12217769 and rs745480) that are associated with macular thickness and refractive error in human. Whereas this gene was not functionally characterized at the time our screen was conducted, a recent study showed that *lrit2* knockdown in zebrafish led to a reduction in eye size (*Chiang et al., 2020*). Finally, a high-throughput screen in mouse found that *Pkd1l2* KO leads to abnormalities in lens morphology (*Dickinson et al., 2016*), but no follow-up experiments exist. A potential role of this gene in human disease is also indicated by an intronic SNP (rs873693), which is associated with age onset of myopia (GWAS catalog, *Hysi et al., 2020*; *Tedja et al., 2018*). Thus, in addition to *SERPINE3*, for which we performed an initial functional characterization in zebrafish, our screen uncovered other promising, uncharacterized candidates that may have an eye-related function. Overall, this highlights the potential of comparative genomics to shed light on the functional roles of less characterized genes and to help to further identify human disease-causing genes (*Meadows and Lindblad-Toh, 2017*).

## Materials and methods

### Visual acuity values

We used publicly available visual acuity measurements (*Veilleux and Kirk, 2014*; *Kemp and Kirk, 2014*; *Figure 1—source data 1* lists all primary references) to classify placental mammals into visual high-acuity and low-acuity species. Visual acuity can be measured by either behavioral experiments or calculated from the eye axial diameter, the peak ganglion cell density and a correction factor for diel activity (*Veilleux and Kirk, 2014*). For species for which both measures were available, we used the behavioral visual acuity as this measure is more accurate (*Veilleux and Kirk, 2014*). For species that lack visual acuity data, we used available visual acuity measurements of closely related species of the same genus or family (*Figure 1—source data 1*). Three subterranean mammals lack available visual acuity measurements (cape golden mole, star-nosed mole, blind mole rat) but exhibit highly degenerated eyes. We therefore assumed a visual acuity of zero. In total, visual acuity values were obtained for 49 placental mammals that were included in a previously generated whole genome alignment (*Sharma and Hiller, 2017*). Using a visual acuity threshold of one, we considered ten mammals (cape golden mole, naked mole rat, star-nosed mole, blind mole rat, big brown bat, little brown bat, David's myotis bat, mouse, prairie vole, deer mouse) representing seven lineages, as low-acuity vision species. All other 39 species with visual acuity greater than one were considered as high-acuity vision species.

### Forward genomics screen

To screen for genes that are preferentially lost in low-acuity placental mammals, we used a previously generated data set of inactivated genes (*Sharma et al., 2018*). Briefly, gene losses were detected with a pipeline that searches for gene-inactivating mutations and performs a number of filtering steps to distinguish between real mutations and artifacts related to genome assembly or alignment issues and exon-intron structure changes. A genome alignment with human (hg38 assembly) as the reference (*Sharma and Hiller, 2017*), human genes annotated by Ensembl (version 87) (*Aken et al., 2017*) and principal isoforms from the APPRIS database (*Rodriguez et al., 2018*) were used as input. Based on the relative positions of inactivating mutations, a value measuring the maximum percent of the reading frame that remains intact (%intact) was computed for each gene and species (*Figure 1—source data 8*). A gene was classified as lost if %intact was <60%. A gene was classified as intact if %intact was ≥90%.

Using the human Ensembl version 87 gene annotation, we excluded protein-coding genes that have missing data due to assembly incompleteness for more than 50% of low- or high-acuity species and genes that had identical %intact values across species, since such genes cannot show associations with lower visual acuity. This resulted in 13,172 genes. *Figure 1—figure supplement 1* provides a flowchart.

To search for genes that tend to have lower %intact values in the low-acuity group, we applied the Forward Genomics approach (*Prudent et al., 2016*; *Sharma et al., 2018*) with a phenotype definition based on visual acuity to these 13,172 genes. We used phylogenetic generalized least squares (*Freckleton et al., 2002*) to account for phylogenetic relatedness and ranked genes by the Benjamini-Hochberg corrected p-value (FDR <0.05). The list of p-values for all genes is provided in *Figure 1—source data 2*. The resulting set of 68 genes is enriched in the GO-terms 'visual perception (GO:0007601)' and 'sensory perception of light stimulus (GO:0050953)' (*Figure 1—source data 9A*). Finally, to obtain candidates that are convergently lost, we required that a gene was lost in at least three of the seven independent low-acuity lineages. This filtering step removed 37 genes and these genes are not enriched in any GO-term (*Figure 1—source data 9B*). Of the remaining 31 genes, two genes are no longer contained in the latest Ensembl annotation (version 106) and were therefore dropped. The other 29 genes are presented in *Figure 1* and considered in the enrichment analysis (below).

To test whether the identification of *SERPINE3*, which we experimentally investigated, is robust to the selected visual acuity threshold, we re-ran the screen after increasing or decreasing the threshold while keeping other parameters constant. Using a more inclusive definition of low-acuity species by increasing the visual acuity threshold to 1.5, which additionally considered manatee, the two flying foxes and rats as low-acuity species, identified a total of seven genes at an FDR of 0.05, with *SERPINE3* at the first rank (*Figure 1—source data 5*). Using a more restrictive definition of low-acuity species by

decreasing the visual acuity threshold to 0.5 identified 31 genes at an FDR of 0.05 with *SERPINE3* at rank 8 (*Figure 1—source data 6*).

In addition to using phenotype definitions based on visual acuity values, we also tested a phenotype definition based on a molecular signature of losses of known vision-related genes. To this end, we ranked all 49 species based on their number of lost genes annotated with the GO-term 'visual perception' (total of 93 genes, *Figure 1—source data 1*). The set of ten species that have lost five or more visual perception genes is similar to the set of low-acuity species used in the original screen. The differences are that mouse and deer mouse are substituted by manatee and goat. Re-running the screen based on this molecular loss pattern of vision genes, retrieved *SERPINE3* at rank 5 (*Figure 1—source data 7*).

As a control to ensure that a Forward Genomics screen does not always retrieve vision-related genes, we ran a new screen, searching for genes preferentially lost in high-acuity sister species (elephant, rhinoceros, horse, two flying foxes, guinea pig, degu, squirrel) of the low-acuity mammals that we used in the original screen. All other species including the other high-acuity mammals were then treated as background (*Figure 1—source data 4*).

## Enrichment analysis

Enrichment analysis was performed using the Enrichr web service (*Kuleshov et al., 2016*), which uses a two-sided Fisher's exact test and corrects for multiple testing with the Benjamini-Hochberg method.

## Investigating and validating *SERPINE3* losses in additional genomes

To explore conservation and loss of *SERPINE3* in additional mammalian genomes that became available since our initial screen, we used the TOGA method (Tool to infer Orthologs from Genome Alignments; Kirilenko et al., titled 'Integrating gene annotation with orthology inference at scale', under review). TOGA uses pairwise alignments between a reference (here human hg38) and a query genome, infers orthologous loci of a gene with a machine learning approach, and uses CESAR 2.0 (*Sharma et al., 2017*) to align the exons of the reference gene to the orthologous locus in the query. TOGA then classifies each transcript by determining whether the central 80% of the transcript's coding sequence encodes an intact reading frame (classified as intact) or exhibits at least one gene-inactivating mutation (classified as potentially lost). If less than 50% of the coding sequence is present in the assembly, the transcript is classified as missing.

Focusing on the evolutionarily conserved human *SERPINE3* Ensembl (version 104) transcript ENST00000524365 (*Howe et al., 2021*), we analyzed the TOGA transcript classification for 418 assemblies that were not used in the initial screen (*Figure 2—source data 1*). These assemblies represent 381 new placental mammal species. For species, where TOGA classified the *SERPINE3* transcript as missing, we inspected the orthologous alignment chain to distinguish intact and lost orthologs from truly missing orthologs due to assembly gaps. Since assembly base errors can mimic false gene losses (*Hecker et al., 2019b*; *Sharma et al., 2020*), we validated inactivating mutations via two approaches. We considered a mutation as real if it is either shared with an independent assembly of the same or a closely-related species, or if alignments of raw DNA sequencing reads support the mutation. Support by raw DNA reads was assessed by aligning the genomic sequence around the mutation against the NCBI short read archive (SRA queried via NCBI megablast) (*Leinonen, 2011*; *Figure 2—source data 2*). With the exception of okapi, steenbok and fox, where *SERPINE3* does not evolve under relaxed selection (*Figure 2—source data 4*), we considered a gene loss as real if there is at least one validated inactivating mutation. We did not validate *SERPINE3* mutations in the genomes of cyclops roundleaf bat and Stoliczka's trident bat, as both species have each eight mutations across several exons and their PacBio HiFi read-based assemblies have very high base accuracies (QV >60, indicating less than one base error per 1 Mb). In order to map *SERPINE3* loss events on the phylogenetic tree, we searched for gene-inactivating mutations that are shared among phylogenetically related species, where parsimony indicates that these mutations and thus gene loss likely occurred in their common ancestor. Those mutations are shown in boxes in *Figure 2—figure supplements 2–8*, all other mutations are the output of CESAR.

To explore if *SERPINE3* remnants are still expressed, we mined expression for eleven *SERPINE3*-loss species by blasting the TOGA transcript annotation against publicly-available RNA-Seq data of

eye or brain tissue. We counted the number of reads that have at least 95% identity (*Figure 2—source data 3*).

## Selection analysis

For species with an unclear *SERPINE3* loss status (mole vole, steenbok, okapi, fox, Steller's sea cow) and species or clades that have an intact *SERPINE3* but many close relatives have lost the gene (greater sac-winged bat, European hedgehog, elephants), we tested whether *SERPINE3* evolves under relaxed selection. To this end, we used RELAX from the HyPhy suite (*Wertheim et al., 2015*) to test whether selection pressure was relaxed (selection intensity parameter K<1) or intensified (K>1) in this species or clade, which we labeled as foreground (*Figure 2—source data 4*). We restricted the analysis to 327 *SERPINE3* that are intact and complete (middle 80% of the coding sequence present) and treated those as background. Codon sequences were obtained from TOGA and aligned with MACSE v2.0 (*Ranwez et al., 2018*). This procedure was repeated using only one foreground species/clade at the time. The species tree used for the analysis is provided as *Figure 2—source data 4*.

## Protein sequence analysis and structure prediction

Signal peptides and the cellular location were predicted with the SignalP 5.0 webserver (*Almagro Armenteros et al., 2019*) and DeepLoc 1 (*Almagro Armenteros et al., 2017*), respectively, for all intact and complete SERPINE3 protein sequences as defined above. Protein sequences were aligned with muscle (*Edgar, 2004*) and visualized with Jalview (*Waterhouse et al., 2009*; *Supplementary file 1*).

The three-dimensional structure of human SERPINE3 was retrieved from the AlphaFold2 web server (*Jumper et al., 2021*; *Supplementary file 2*). We calculated the root mean square distance (RMSD) to all homologous chains of existing crystal structures of close serpin relatives in native state (*Supplementary file 3*) after structural alignment in PyMOL (*Schrödinger and DeLano, 2020*). For each crystal structure, we averaged the RMSD for all chains.

## Mammalian SERPINE3 have features of inhibitory, secreted SERPINs

Mammalian SERPINE3 carries an N-terminal signal peptide (*Figure 2—figure supplement 9*) that is predicted to lead to their secretion into the extracellular space in 96% of all analyzed intact and complete SERPINE3. SignalP 5.0 did not predict the presence of a signal peptide for three species (probability<50%): fat dormouse, black flying fox and puma.

Sequence analysis of intact and complete SERPINE3s revealed that two key features of inhibitory serpins are well conserved among placental mammals (sequence alignment is provided as *Supplementary file 1*). First, the substrate determining residues P4-P4' are conserved (positions 366–372 in human SERPINE3), whereby P1 denotes the substrate binding scissile bond, position 369 (*Figure 2—figure supplement 10*; *Khan et al., 2011*). This position is occupied by an arginine in SERPINE3 as is the case in the inhibitory SERPINE1 and SERPINE2 proteins. Second, the close-by hinge region (positions 355–361) is mostly occupied by small amino acids without prolines. This may allow the insertion of the hinge region into the A beta sheet, a key feature of serpins' inhibitory mechanism (*Simonovic et al., 2001*).

Furthermore, AlphaFold2 (*Jumper et al., 2021*) predicted a three-dimensional structure of the human SERPINE3 (*Supplementary file 2*) that is very similar to other native serpins (mean RMSD to native structures 1.6A, *Supplementary file 3*) and adopts the native fold of serpins with an exposed, disordered reactive core loop for substrate binding that does not seem to adopt an alpha-helical conformation as in the non-inhibitory ovalbumin (*Stein et al., 1991*). Taken together, this suggests that SERPINE3 functions as a secreted serine protease inhibitor.

## Mining gene function, expression, and genetic variation sources

Information on the function of genes discovered in our screen was obtained from GeneCards database (*Stelzer et al., 2016*), UniProt (*UniProtConsortium, 2021*), Ensembl (*Howe et al., 2021*), Proteomics DB (*Samaras et al., 2020*), the Human protein atlas (*Uhlén et al., 2019*), the Expression atlas and Single cell expression atlas (*Papatheodorou et al., 2020*). Expression in human eye for each candidate gene were obtained by averaging expression over all healthy, primary RNA-Seq data sets per tissue (cornea, RPE, retina) provided by the eyeIntegration database (*Bryan et al., 2018*). Cell lines

were not included in the average. A gene was considered to be expressed if the Transcripts Per Million (TPM) value was >100 in cornea, RPE, or retina. Expression of *SERPINE3* was further assessed by retrieving primary data sets from FantomCat (*Hon et al., 2017*), GEO profiles (*Barrett et al., 2013*), and Bgee (*Bastian et al., 2021*). *Figure 3—source data 5* provides the list of all data sets.

Phenotype associations of SNPs located in loci of interest were investigated based on the GWAS catalog (*MacArthur et al., 2017*), dbSNP (*Sherry et al., 2001*) and PheGenI (*Ramos et al., 2014*). Linkage of SNPs with a candidate gene in 30 human populations was investigated based on the GWAS catalog. To evaluate possible functional consequences of the respective SNPs, we overlapped their (projected) coordinates with regulatory elements from ENCODE for human (hg38) and mouse (mm10) via the web-based server SCREEN v. 2020–10 (*Moore et al., 2020*). Additionally, we investigated eye- and retina-associated regulatory elements in the Ensembl and UCSC genome browsers (*Lee et al., 2022*; *Howe et al., 2021*).

## Animal husbandry

Adult zebrafish (*Danio rerio*, AB line) were maintained at 26.5 °C with a 10/14 hr dark/light cycle (*Brand and Granato, 2002*). Embryos and larvae were raised at 28.5 °C in the dark until six days old. For phenotyping, we used adult fish of both *serpine3* KO lines (*serpine3*[cbg17]:19 months, *serpine3*[cbg18]: 11 months) generated in this study as well as their WT siblings. WT mice (*Mus musculus*, C57BL/6JOlaHsd line) were maintained in a barrier system at 20°C–24°C with a 12/12 hr dark/light cycle.

## Expression analysis by RT-qPCR

Adult zebrafish >12 months were sacrificed by rapid cooling after anesthesia with MESAB. Adult mice (2 months, male) were sacrificed by cranial dislocation after carbon dioxide anesthesia. Tissues were dissected in ice-cold phosphate buffer (PBS), frozen in liquid nitrogen and stored at –80 °C until further use. RNA was extracted from lysed, homogenized tissue with RNeasy mini or midi kits (Qiagen) according to manufacturer's instructions and reprecipitated if necessary. The RNA Integrity Number (RIN) was >7 for all tissues except spleen (RIN >6). Intact total RNA was reverse transcribed into cDNA using the ProtoScript II First Strand cDNA Synthesis Kit (NEB) according to manufacturer's instructions with random primers. RT-qPCR was performed after addition of SybrGreen (Roche). Expression relative to a normalization gene was calculated from Ct values according to the efficiency and delta delta Ct method. Specifically, relative ratios of *Serpine3*/*serpine3* expression (Zf_serp_F: GAGACCCAAAACCTGCCCTT, Zf_serp_R: AGCCGGAAATGACCGATATTGA, Mm_serp_F: TGGAGCTTTCAGAGGAGGGTA, Mm_serp_R: GATACTGAAGACAAACCCTGTGC) were obtained by using *rpl13a* (Zf_rpl13a_F: TCTGGAGGAACTGTAAGAGGTATGC, Zf_rpl13a_R: AGACGCACAATCTTGAGAGCGA) or *actb* (Zf_actb_F: CGAGCAGGAGATGGGAACC, Zf_actb_R: CAACGGAAACGCTCATTGC) as the reference gene for zebrafish and *Rpl27* as the reference gene for mouse (Mm_Rpl27_F: TTGAGGAGCGATACAAGACAGG, Mm_Rpl27_R: CCCAGTCTCTTCCCACACAAA). At least three biological replicates per sample group were analyzed, which were each represented by the average normalized relative ratio of three to six technical replicates. For zebrafish eye with *actb* as the normalizer, one biological replicate was an expression outlier for eye and liver, defined as exceeding the third quartile by more than three times the interquartile distance (Q3 +3*interquartile distance). These two data points were excluded from *Figure 3A* but are contained in *Figure 3—source data 1*.

## ISH and FISH

For ISH and immunostainings, fish were sacrificed, eyes dissected and fixed in 4% paraformaldehyde/0.1 M PBS after removal of the lens. Eyes were embedded in gelatin/sucrose and sectioned (14 µm) with a cryostat. The *serpine3* in situ probe spans the coding exons 3–8 (transcript: ENSDART00000132915.2). Using primers with restriction enzyme cut sites (forward, Not1: TAAGCA GC GGCCGCGTAAAAGT GCCCATGATGTACCAG, reverse, BamHI: TAAGCA G GATCC ACAACTCGACCTATAAACAGCAAC), *serpine3* cDNA was amplified from total cDNA and cloned into the pCRII-topo vector (Invitrogen). The antisense probes were transcribed with SP6 polymerase, using a DIG-labeled NTP mix (Roche diagnostics). The (fluorescent) ISH were conducted as previously described with minor modifications (*de Oliveira-Carlos et al., 2013*): Hybridization and washing steps were performed at 60 °C. For chromogenic in situs, sections were incubated with anti-digoxigenin-AP (Roche), diluted 1:4000 in DIG-blocking reagent (Roche) at 4 °C overnight and subsequently developed with NBT/BCIP (Roche). For

FISH, sections were washed in PBS immediately after quenching. Sections were blocked for 1 hr with 2% blocking reagent in MABT (Perkin-Elmer) and then incubated with anti-digoxigenin-POD (Roche), diluted 1:500. The signal was detected with the TSA Plus Cy3/Cy5 kit (Perkin-Elmer).

## Immunohistochemistry

Immunostainings for glial fibrillary acidic protein (ZRF1 from DSHB, 1:200), choline O-acetyltransferase (AB144P from Millipore, 1:500) and protein kinase C alpha (SC-208 from Santa-Cruz, 1:500) were performed after completion of the FISH protocol according to *de Oliveira-Carlos et al., 2013*. For chat, antigens were retrieved in preheated 10 mM sodium citrate buffer for 6' at 85 °C. Sections were washed in PBS and 0.3% PBSTx prior to primary antibody incubation. Following the protocol of *de Oliveira-Carlos et al., 2013*, we washed the sections three times in PBSTx, and incubated in anti-goat or anti-rabbit IgG (H+L) Alexa 488-conjugated secondary antibodies (Invitrogen, 1:750).

## Generation of KO lines and genotyping

To generate *serpine3* KO lines, deletions were introduced using the CRISPR-Cas9 system. Guides were chosen considering efficiency predictions of the IDT DNA CRISPR-Cas9 guide checker and Chop-ChopV2 (*Labun et al., 2016*) on the zebrafish assembly danRer7 and ordered as Alt-R CRISPR-Cas crRNA from IDT DNA. For each line, three guides were simultaneously injected into one-cell stage zebrafish embryos as ribonucleoprotein delivery using the Alt-R CRISPR-Cas system following the manufacturer's instructions (0.5 fmol crRNA per embryo per guide, 0.68 ng Cas9 protein per embryo). The expected deletions were confirmed by Sanger sequencing in several founder individuals, one of which was chosen as the founder of each line (*Figure 4—figure supplement 1*). Heterozygous cbg17 and cbg18 zebrafish were further outcrossed to WT fish for several generations and then bred to homozygosity.

More specifically, for s*erpine3*[cbg17], we abolished *serpine3* transcription by deleting the single transcription start site, which is supported by activating histone marks in zebrafish and is also well conserved in human and mouse (*Figure 4—figure supplement 1A*), using the following guides: GGTA TTTGTACTCTAATGAA (guide1), TGTACTCTAATGAAAGGAAC (guide2), CTCACACAGGACAATC CGGCAGG (guide3). For genotyping, we used primers (For1: 5-GAAATCGCATGTCACGCAGAAAT-3, Rev2: 5-ATATCGGAACTGACATACTGAACG-3, Rev2.2: 5- GTGAGCTTCGTGTTTGTGGT-3) to amplify a region around the transcription start site (*Figure 4—figure supplement 2A, B*, *Figure 4—source data 4 and 5*).

*Serpine3*[cbg18] was generated by introducing a frame-shifting deletion in coding exon 1, which presumably results in three early stop codons when the transcript is translated. The following guides were used: TCTTCTGCAACTCGGGGCCA (guide4), TCTCTGTGAGCGTCTGGTAG (guide5), AACA CTCTGGTTCAGCTCGA (guide6) (*Figure 4—figure supplement 1B*). We genotyped fish by ampli-fying a region around coding exon1 with the following primers: For3: 5-GGCATTGTTGAGATTCA GTAGTCA-3, Rev4: 5-CAGTTTACTCCTACCATTGACATC-3 (*Figure 4—figure supplement 2C*, *Figure 4—source data 6*).

## Histology

For hematoxylin/eosin stainings, fish were sacrificed and heads were fixed overnight at 4 °C in 4% paraformaldehyde/ 0.1 M PBS and decalcified in 0.5 M EDTA in 0.1 M PBS for 3–4 days. Next, they were processed in a Paraffin-Infiltration-Processor (STP 420, Zeiss) according to the following program: ddH$_2$0: 1×1'; 50% ethanol (EtOH) 1×5'; 70% EtOH 1×10'; 96% EtOH 1×25'; 96% EtOH 2×20'; 100% EtOH 2×20'; xylene 2×20'; paraffin 3×40'/60 °C; paraffin 1×60'/60 °C. The heads were embedded in paraffin using the Embedding Center EG1160 (Leica). Semi-thin sections (2 µm) were cut on an Ultracut microtome (Mikrom) and counterstained using hematoxylin/eosin (HE, Sigma).

## Microscopy, image processing, and analysis

Imaging was performed using the ZEISS Axio Imager.Z1 provided by the CMCB Light Microscopy Facility. The images were processed in Fiji/ImageJ version 2.1.0 (*Schindelin et al., 2012*) (macroscopic eye images) and Adobe Illustrator. For macroscopic phenotyping, eyes were imaged with a Leica stereo microscope M165C. To parameterize the eye shape, the eye outline was first approximated by an oval and then manually corrected if necessary. Particle parameters of the final eye object were

measured automatically in Fiji. The statistical analysis and visualization were conducted in R version 4.1.0 (2021-05-18) using the packages ggplot2 (*Wickham, 2016*), tseries (*Trapletti and Hornik, 2021*) and effsize. Outliers were not excluded. Comparing WT and KO individuals, we tested whether both genotypes have the same iris shape (estimated by iris solidity and circularity) using the Wilcoxon rank sum test after rejecting normality of the variables with a Jarque Bera Test. Both eyes of the same individual were treated as individual biological replicates, since we observed shape deviations often only in one eye of the same individual (*Figure 4—source data 2*).

### Animal licenses

All experiments in mouse and zebrafish were performed in accordance with the German animal welfare legislation. Protocols were approved by the Institutional Animal Welfare Officer (Tierschutzbeauftragter), and licensed by the regional Ethical Commission for Animal Experimentation (Landesdirektion Sachsen, Germany; license no. DD24-5131/354/11, DD24.1-5131/451/8, DD24-5131/346/11, DD24-5131/346/12).

### Acknowledgements

We thank the genomics community for sequencing and assembling the genomes and the UCSC genome browser group for providing software and genome annotations. Experimental work would not have been possible (or as pleasant) without supporting hosting labs, especially Nadine Vastenhouw, Elisabeth Knust and Wieland Huttner. We also thank Nadine Vastenhouw and her whole lab, Michael Heide and Mauricio Rocha as well as current and former members of the Hiller lab for helpful scientific discussion and comments on the manuscript. We thank the following facilities of MPI-CBG: Biomedical Services (especially fish unit), Cell technologies (Julia Jarrells), Sequencing and genotyping (Sylke Winkler), Light Microscopy, Scientific Computing, Computer Service Facilities as well as the Computer Service Facilities of MPI-PKS and the CMCB Histology (Susanne Weiche) and CMCB Light Microscopy Facility for their support. Work of HI and MH was supported by an exploration grant from the Boehringer Ingelheim Stiftung, the Max Planck Society and the LOEWE Centre for Translational Biodiversity Genomics (TBG) funded by the Hessen State Ministry of Higher Education, Research and the Arts (HMWK). Work by JH, AM, SH and MB was supported by project grants of the German Research Foundation (Deutsche Forschungsgemeinschaft, project numbers BR 1746/3 and BR 1746/6) and an ERC advanced grant (Zf-BrainReg) to MB. This study was furthermore supported with a PhD scholarship from Studienstiftung des deutschen Volkes to JH.

## Additional information

### Funding

| Funder | Grant reference number | Author |
|---|---|---|
| Boehringer Ingelheim | | Michael Hiller |
| Max Planck Society | | Michael Hiller |
| Hessen State Ministry of Higher Education, Research and the Arts | | Michael Hiller |
| German Research Foundation | BR 1746/3 and BR 1746/6 | Michael Brand |
| European Research Council | Zf-BrainReg | Michael Brand |
| Studienstiftung des Deutschen Volkes | | Juliane Hammer |

The funders had no role in study design, data collection and interpretation, or the decision to submit the work for publication.

## Author contributions
Henrike Indrischek, Data curation, Formal analysis, Investigation, Methodology, Resources, Visualization, Writing – original draft, Writing – review and editing; Juliane Hammer, Anja Machate, Formal analysis, Investigation, Resources, Visualization, Writing – review and editing; Nikolai Hecker, Conceptualization, Formal analysis, Investigation, Methodology, Resources, Writing – review and editing; Bogdan Kirilenko, Investigation, Resources, Software, Writing – review and editing; Juliana Roscito, Formal analysis, Investigation, Resources, Writing – review and editing; Stefan Hans, Formal analysis, Investigation, Methodology, Resources, Writing – review and editing; Caren Norden, Investigation, Methodology, Resources, Writing – review and editing; Michael Brand, Conceptualization, Funding acquisition, Methodology, Project administration, Resources, Supervision, Writing – review and editing; Michael Hiller, Conceptualization, Funding acquisition, Investigation, Project administration, Supervision, Visualization, Writing – original draft, Writing – review and editing

## Author ORCIDs
Henrike Indrischek (ID) http://orcid.org/0000-0002-2810-5272
Juliane Hammer (ID) http://orcid.org/0000-0003-2511-9537
Nikolai Hecker (ID) http://orcid.org/0000-0003-1693-4257
Caren Norden (ID) http://orcid.org/0000-0001-8835-1451
Michael Brand (ID) http://orcid.org/0000-0001-5711-6512
Michael Hiller (ID) http://orcid.org/0000-0003-3024-1449

## Ethics
All experiments in mouse and zebrafish were performed in accordance with the German animal welfare legislation. Protocols were approved by the Institutional Animal Welfare Officer (Tierschutzbeauftragter), and licensed by the regional Ethical Commission for Animal Experimentation (Landesdirektion Sachsen, Germany; license no. DD24-5131/354/11, DD24.1-5131/451/8, DD24-5131/346/11, DD24-5131/346/12).

## Decision letter and Author response
Decision letter https://doi.org/10.7554/eLife.77999.sa1
Author response https://doi.org/10.7554/eLife.77999.sa2

---

# Additional files

## Supplementary files
• Supplementary file 1. Annotated protein alignment of intact and complete mammalian SERPINE3. jvp format is to be opened in Jalview.

• Supplementary file 2. Predicted structure of human SERPINE3 (AlphaFold 2).

• Supplementary file 3. Comparison of 3D structures of crystallized SERPINs to the predicted human SERPINE3.

• Transparent reporting form

## Data availability
All data needed to evaluate the conclusions in the paper are present in the paper and the Supplementary Materials. The annotated protein alignment of intact and complete mammalian SERPINE3 genes are in Supplementary File 1, the predicted structure of human SERPINE3 are in Supplementary File 2 and the raw microscopy images of fish eyes are available on Dryad at DOI 10.5061/dryad.ncjsxksxt. TOGA annotations of SERPINE3 and a visualization of gene-inactivating mutations are available at our UCSC genome browser mirror https://genome.senckenberg.de.

The following dataset was generated:

| Author(s) | Year | Dataset title | Dataset URL | Database and Identifier |
|---|---|---|---|---|
| Indrischek H, Hiller M, Brand M | 2022 | Microscopic eye phenotype of SERPINE3 cbg17 and cbg18 Knockout lines | https://doi.org/10.5061/dryad.ncjsxksxt | Dryad Digital Repository, 10.5061/dryad.ncjsxksxt |

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

## Appendix 1

### Appendix 1—key resources table

| Reagent type (species) or resource | Designation | Source or reference | Identifiers | Additional information |
|---|---|---|---|---|
| gene (*Danio rerio*) | *serpine3* | Ensembl | ENSDART00000132915.2 | |
| gene (*Homo sapiens*) | *SERPINE3* | Ensembl | ENST00000524365 | |
| strain, strain background (*Danio rerio*) | WT (AB) | Zebrafish International Resource Center | | |
| strain, strain background (*Mus musculus*) | WT (C57BL/6JOlaHsd) | Envigio | | |
| genetic reagent (*Danio rerio*) | cbg17 | This paper | | See Materials and Methods "Generation of KO lines and genotyping" |
| genetic reagent (*Danio rerio*) | cbg18 | This paper | | See Materials and Methods "Generation of KO lines and genotyping" |
| Antibody | Anti-digoxigenin-AP (polyclonal sheep) | Sigma-Aldrich | Cat. #: 11093274910, RRID: AB_2734716 | IF (1:4000) |
| Antibody | Anti-digoxigenin-POD conjugated (polyclonal sheep) | Roche | Cat. #: 11207733910, RRID: AB_514500 | IF (1:500) |
| Antibody | Anti-gfap ZRF-1 (monoclonal mouse) | Zebrafish International Resource Center | Cat. #: zrf-1, RRID: AB_10013806 | IF (1:200) |
| antibody | Anti-choline acetyltransferase (polyclonal goat) | Sigma-Aldrich | Millipore Cat. #: AB144P, RRID: AB_2079751 | IF (1:500) |
| antibody | Anti-protein kinase C alpha (polyclonal rabbit) | Santa Cruz Biotechnology | Cat. #: sc-208, RRID: AB_2168668 | IF (1:500) |
| antibody | Anti-goat IgG (H+L) Alexa 488 (polyclonal chicken) | Invitrogen | Cat. #: A21467 | IF (1:750) |
| antibody | Anti-rabbit IgG (H+L) Alexa 488 (polyclonal goat) | Invitrogen | Cat. #: A11034 | IF (1:750) |
| recombinant DNA reagent | serpine3-pCRII-topo vector | This paper | | See Materials and Methods "Generation of KO lines and genotyping" |
| sequence-based reagent | Zf_serp_F | This paper | qPCR primer | GAGACCCAAAACCTGCCCTT |
| sequence-based reagent | Zf_serp_R | This paper | qPCR primer | AGCCGGAAATGACCGATATTGA |
| sequence-based reagent | Mm_serp_F | This paper | qPCR primer | TGGAGCTTTCAGAGGAGGGTA |
| sequence-based reagent | Mm_serp_R | This paper | qPCR primer | GATACTGAAGACAAACCCTGTGC |
| sequence-based reagent | Zf_rpl13a_F | https://doi.org/10.1111/j.1745-7270.2007.00283.x | qPCR primer | TCTGGAGGAACTGTAAGAGGTATGC |
| sequence-based reagent | Zf_rpl13a_R | https://doi.org/10.1111/j.1745-7270.2007.00283.x | qPCR primer | AGACGCACAATCTTGAGAGCGA |
| sequence-based reagent | Zf_actb_F | https://doi.org/10.1007/s00441-020-03318-2 | qPCR primer | CGAGCAGGAGATGGGAACC |
| sequence-based reagent | Zf_actb_R | https://doi.org/10.1007/s00441-020-03318-2 | qPCR primer | CAACGGAAACGCTCATTGC |

*Appendix 1 Continued on next page*

*Appendix 1 Continued*

| Reagent type (species) or resource | Designation | Source or reference | Identifiers | Additional information |
|---|---|---|---|---|
| sequence-based reagent | Mm_Rpl27_F | This paper | qPCR primer | TTGAGGAGCGATACAAGACAGG |
| sequence-based reagent | Mm_Rpl27_R | This paper | qPCR primer | CCCAGTCTCTTCCCACACAAA |
| sequence-based reagent | Guide1 | This paper | Crispr guide | GGTATTTGTACTCTAATGAA |
| sequence-based reagent | Guide2 | This paper | Crispr guide | TGTACTCTAATGAAAGGAAC |
| sequence-based reagent | Guide3 | This paper | Crispr guide | CTCACACAGGACAATCCGGCAGG |
| sequence-based reagent | Guide4 | This paper | Crispr guide | TCTTCTGCAACTCGGGGCCA |
| sequence-based reagent | Guide5 | This paper | Crispr guide | TCTCTGTGAGCGTCTGGTAG |
| sequence-based reagent | Guide6 | This paper | Crispr guide | AACACTCTGGTTCAGCTCGA |
| sequence-based reagent | For1 | This paper | Genotyping primer | GAAATCGCATGTCACGCAGAAAT |
| sequence-based reagent | Rev2 | This paper | Genotyping primer | ATATCGGAACTGACATACTGAACG |
| sequence-based reagent | Rev2.2 | This paper | Genotyping primer | GTGAGCTTCGTGTTTGTGGT |
| sequence-based reagent | For3 | This paper | Genotyping primer | GGCATTGTTGAGATTCAGTAGTCA |
| sequence-based reagent | Rev4 | This paper | Genotyping primer | CAGTTTACTCCTACCATTGACATC |
| peptide, recombinant protein | Alt-R CRISPR-Cas 9 Nuclease | IDT | Cat. #: 10000735 | |
| commercial assay or kit | RNeasy mini kit | Qiagen | Cat. #: 74,106 | |
| commercial assay or kit | RNeasy midi kit | Qiagen | Cat. #: 75,144 | |
| commercial assay or kit | ProtoScript II First Strand Synthesis kit | NEB | Cat. #: E6560S | |
| commercial assay or kit | TSA Plus Cy3/Cy5 kit | Perkin-Elmer | Cat. #: NEL744001KT | |
| commercial assay or kit | Alt-R CRISPR-Cas9 tracr RNA | IDT | Cat. #: 1072532 | |
| software, algorithm | Enrichr | https://doi.org/10.1093/nar/gkw377 | RRID:SCR_001575 | |
| software, algorithm | TOGA | Kirilenko et al., titled 'Integrating gene annotation with orthology inference at scale', under review (https://github.com/hillerlab/TOGA; *Kirilenko and Hiller, 2022*) | | |
| software, algorithm | CESAR 2.0 | https://doi.org/10.1093/bioinformatics/btx527 | | |
| software, algorithm | RELAX | https://doi.org/10.1093/molbev/msu400 | RRID:SCR_016162 | |

*Appendix 1 Continued on next page*

*Appendix 1 Continued*

| Reagent type (species) or resource | Designation | Source or reference | Identifiers | Additional information |
|---|---|---|---|---|
| software, algorithm | MACSE v2.0 | https://doi.org/10.1093/molbev/msy159 | | |
| software, algorithm | Jalview | https://doi.org/10.1093/molbev/msy159 | RRID:SCR_006459 | |
| software, algorithm | Human protein atlas | https://doi.org/10.1038/nbt1210-1248 | RRID:SCR_006710 | |
| software, algorithm | Gene Expression Atlas | https://doi.org/10.1093/nar/gkz947 | RRID:SCR_007989 | |
| software, algorithm | PyMOL | *Schrödinger and DeLano, 2020* | RRID:SCR_000305 | |
| software, algorithm | Fiji | https://doi.org/10.1038/nmeth.2019 | RRID:SCR_002285 | |
| software, algorithm | R | Other | RRID:SCR_001905 | https://cran.r-project.org/src/base/R-3/ |
| software, algorithm | ggplot2 | Other | RRID:SCR_014601 | http://docs.ggplot2.org/current/ |
| software, algorithm | tseries | Other | | https://CRAN.R-project.org/package=tseries |
| software, algorithm | effsize | Other | | https://cran.r-project.org/web/packages/effsize/index.html |
| software, algorithm | SignalP5 | https://doi.org/10.1038/s41587-019-0036-z | | |
| software, algorithm | DeepLoc | https://doi.org/10.1093/bioinformatics/btx431 | | |
| software, algorithm | ChopChop v2 | https://doi.org/10.1093/nar/gkw398 | RRID:SCR_015723 | |

