## [Editor Report]

The authors use a comparative genomics approach to predict gene function, in particular genes that have a role in eye development. After identifying the convergent loss of SERPINE3 with vision loss across mammals, the authors confirmed its involvement in eye development by characterizing zebrafish knockouts. This work highlights the power of comparative genomics to generate hypotheses that can be experimentally validated. This work is relevant to a broad audience interested in evolution and adaptation as well as for those studying eye development and eye pathologies.

---

## [Decision Letter]

**Decision letter after peer review:**

Thank you for submitting your article "Vision-related convergent gene losses reveal *SERPINE3*'s unknown role in the eye" for consideration by *eLife*. Your article has been reviewed by 2 peer reviewers, and the evaluation has been overseen by a Reviewing Editor and George Perry as the Senior Editor. The following individual involved in review of your submission has agreed to reveal their identity: Stephen Treaster (Reviewer #2).

Essential revisions:

The reviewers were positive but also consistent in their major concerns about the pipeline and its output, requiring more clarity/data and at least some level of additional supporting validation, perhaps including manual curation. I agree that these concerns need to be addressed, including some further cross-checking of the database Hopefully this is possible with existing cross-species RNA-seq datasets, and does not require the generation of new experimental data.

Overall, congratulations on this nice study, and I look forward to seeing the manuscript at its next evolutionary stage!

*Reviewer #1 (Recommendations for the authors):*

I have only one comment: The authors identify a very large number of putative gene losses in many different species. Given that usually the majority of such bioinformatically identified gene losses are false positives due to annotation or sequencing artifacts, a manual curation is usually critical – even with the most sophisticated algorithms. According to the methods this has not been performed. I am aware that the authors are world experts in identifying gene losses, but I am still skeptical that all of these are true events (given the sheer number). I recommend that the authors use existing RNA-Seq data sets (for example from the naked mole rat) or PCR from genomic DNA to confirm at least a subset of the ones that they are showing.

*Reviewer #2 (Recommendations for the authors):*

Many of the filters are explained in the previous publications generating the dataset, however, there seem to be four additional thresholds to reuse that dataset for this analysis. While these filters may all be appropriate, it is opaque to the reader. (1) visual acuity cutoffs, (2) percent of high and low acuity lineages with assembly gaps, (3) requiring an intact score in >80% of high acuity lineages and lost in <10% in high acuity lineages, (4) requiring the gene be lost in at least three of the seven independent low-acuity lineages. There is a brief expansion on filter 1 by trying two other close acuity cutoffs. However, these other acuity cutoffs seem to have dramatic consequences that are not discussed. 20 of the 26 top hits vanish when using a cutoff of 2 instead of 1. With such an impact, it is difficult to interpret what exactly is being measured or what the "right" cutoff may be to get the best measure of vision loss. This is exacerbated by the unintuitive nature of visual acuity scores. Unlike the iconic convergence examples, the operational difference between visual acuity scores of 0.5, 1, and 2, or even 2 and 20, is not obvious. If a continuous measure is going to be binarized, there should be a more apparent argument for the cutoffs, particularly since the 0.5 cutoff has more significant results than 1, but is not the focus of the manuscript.

Filters 3 and 4 appear to be redundant with each other and convergence analysis itself. The analysis is already scoring genes for intactness in conjunction with the trait, so it seems unnecessary to exclude genes from the analysis based on patterns of intactness in conjunction with the trait. It's the same measurement, twice. The purpose of these filters is unclear. The consequences of these filters on p-value distributions, multiple hypothesis corrections, and top hits are unclear, and possibly inappropriate. The lack of a negative control again exacerbates these concerns. While it is non-trivial to select a control set (or many sets) of lineages with comparable topological relationships as the experimental group, it is necessary to demonstrate the robustness of the method and dataset.

Including entire gene lists in the supplemental tables for each method (as opposed to just the top 26 of a single method) would helpfully provide the raw p-values for the distributions mentioned above, allow more comparison and quantification of differences between cutoffs, and allow for future intersections with other datasets. This would dramatically increase the utility of the overall convergence results for the field. In conjunction, there is a truly impressive amount of genomic work here that is unlikely to be replicated by someone else that wants to do a similar analysis on a different phenotype, but the dataset itself does not seem to be available.

---

## [Author Response]

Essential revisions:The reviewers were positive but also consistent in their major concerns about the pipeline and its output, requiring more clarity/data and at least some level of additional supporting validation, perhaps including manual curation. I agree that these concerns need to be addressed, including some further cross-checking of the database Hopefully this is possible with existing cross-species RNA-seq datasets, and does not require the generation of new experimental data.Overall, congratulations on this nice study, and I look forward to seeing the manuscript at its next evolutionary stage!

We thank the editor for these encouraging words.

As detailed below, we have analyzed publicly-available RNA-seq data of different species and find in all cases no relevant expression of the remnants of the inactivated *SERPINE3* gene. We also performed additional validations of the inactivating mutations and every reported *SERPINE3* loss has at least one validated mutation.

Reviewer #1 (Recommendations for the authors):I have only one comment: The authors identify a very large number of putative gene losses in many different species. Given that usually the majority of such bioinformatically identified gene losses are false positives due to annotation or sequencing artifacts, a manual curation is usually critical – even with the most sophisticated algorithms. According to the methods this has not been performed. I am aware that the authors are world experts in identifying gene losses, but I am still skeptical that all of these are true events (given the sheer number). I recommend that the authors use existing RNA-Seq data sets (for example from the naked mole rat) or PCR from genomic DNA to confirm at least a subset of the ones that they are showing.

We apologize for the lack of clarity in the manuscript describing our initial validation of inactivating *SERPINE3* mutations. This description was included in the section “Investigating SERPINE3 in additional genomes”, which is placed downstream of the section describing the screen, since we validated *SERPINE3* not only in the species included in our screen. We have now changed the title of this section to “Investigating and validating *SERPINE3* losses in additional genomes” and rephrased it to improve clarity:

“Since assembly base errors can mimic false gene losses (*32, 85*), we validated inactivating mutations via two approaches. We considered a mutation as real if it is either shared with an independent assembly of the same or a closely-related species, or if alignments of raw DNA sequencing reads support the mutation. Support by raw DNA reads was assessed by aligning the genomic sequence around the mutation against the NCBI short read archive (SRA queried via NCBI megablast) (*86*) (Figure 2 – source data 2). With the exception of okapi, steenbok and fox, where *SERPINE3* does not evolve under relaxed selection (Figure 2 – source data 4), we considered a gene loss as real if there is at least one validated inactivating mutation. We did not validate *SERPINE3* mutations in the genomes of cyclops roundleaf bat and Stoliczka's trident bat, as both species have each eight mutations across several exons and their PacBio HiFi read-based assemblies have very high base accuracies (QV>60, indicating less than one base error per 1 Mb).”

We now also added a sentence about this validation step in the Results section “*SERPINE3* became dispensable in many mammals that do not primarily rely on vision”.

“We only considered *SERPINE3* losses for which at least one inactivating mutation could be validated (see Methods, Figure 2 – source data 1-2). “

We initially validated only mutations for species, where only 1 or 2 mutations in SERPINE3 were observed that are not shared with other assemblies or species. We now extended this analysis to all species, where *SERPINE3* mutations are not shared, irrespective of the number of observed mutations.

With the exception of dikdik and polecat, where putative mutations are likely base errors, we could confirm 28 inactivating mutations in 11 other species.

Details on the validation with raw DNA sequencing reads are provided in the new additional source data “Figure 2 – source data 2” and summarized in “Figure 2 – source data 1”.

I am aware that the authors are world experts in identifying gene losses, but I am still skeptical that all of these are true events (given the sheer number). I recommend that the authors use existing RNA-Seq data sets (for example from the naked mole rat) or PCR from genomic DNA to confirm at least a subset of the ones that they are showing.

As suggested, we used publicly available RNA-seq data to evaluate whether inactivated SERPINE3 genes are still expressed. RNA-seq data of eyes, where *SERPINE3* expression is expected, is available for several bat species. We blasted the full-length *SERPINE3* sequences as annotated by TOGA against these RNA-Seq data sets and obtained a very low number of aligning reads (<10) for all bats. As a positive control, we blasted GLUL, a gene that is expressed in brain and eye, which revealed at least 9000 aligning RNA-seq reads.

For the common vampire bat, we also mapped the entire eye RNA-seq data set against the genome. A new “Figure 2 —figure supplement 1” shows the absence of SERPINE3 expression and the lack of spliced reads that would indicate transcript processing:

For the naked mole rat and other species like blind mole rat, damara mole rat and armadillo, eye RNA-seq data is not available. Therefore, we analyzed available brain RNA-seq data for these species, as our expression tests in zebrafish show that *SERPINE3* is also expressed in the brain. As for eye tissue, we found no indication that remnants of inactivated *SERPINE3* genes are still expressed in the brain.

These data are shown in a new source Data file “Figure 2 – source data 3“ and mentioned in the main text:

“Further supporting *SERPINE3* losses, analyses of publicly-available RNA-seq data indicates that remnants of inactivated *SERPINE3* genes do not show relevant expression (Figure 2 —figure supplement 1, Figure 2 – source data 3). “

Reviewer #2 (Recommendations for the authors):Many of the filters are explained in the previous publications generating the dataset, however, there seem to be four additional thresholds to reuse that dataset for this analysis. While these filters may all be appropriate, it is opaque to the reader. (1) visual acuity cutoffs, (2) percent of high and low acuity lineages with assembly gaps, (3) requiring an intact score in >80% of high acuity lineages and lost in <10% in high acuity lineages, (4) requiring the gene be lost in at least three of the seven independent low-acuity lineages. There is a brief expansion on filter 1 by trying two other close acuity cutoffs. However, these other acuity cutoffs seem to have dramatic consequences that are not discussed.

We thank the reviewer for pointing this out. We addressed this point as follows:

First, we have now simplified the workflow by dropping filter 3. This filter was inspired by previous work, but is not necessary for our vision-related screen as it affects only three genes. This increases the number of genes retrieved from the screen from 26 to 29. The three additional genes are *PKD1L2*, *GJA10*, *LBHD1*, of which *PKD1L2* and *GJA10* have a known eye-related function. These three genes are now included in the updated Figure 1.

Second, we have now added a flowchart as Figure 1 – supplement figure 1, which describes each step and lists how many genes passed each filter.

Third, we explain the rationale behind the first two filters in the revised Methods section. These filters exclude genes that cannot show strong associations because of incomplete data or lack of variation:

“Using the human Ensembl version 87 gene annotation, we excluded protein-coding genes that have missing data due to assembly incompleteness for more than 50% of low- or high-acuity species and genes that had identical %intact values across species, since such genes cannot show associations with lower visual acuity. This resulted in 13,172 genes. Figure 1 —figure supplement 1 provides a flowchart.”

Fourth, we explored the impact of the last filter, which reduced the candidate gene set from 68 to 31. At an FDR of 0.05, we retrieved 68 genes. We now also tested whether this gene set is enriched in eye-related genes. As shown in a new Figure 1 – source data 9A, these 68 genes are significantly enriched in ‘’visual perception” (corrected P-value 9.1E-08) and “sensory perception of light stimulus” (corrected P-value 9.1E-08) functions. The last filter for losses in at least three independent low-acuity lineages attempts to filter out gene losses in ≤2 lineages that are less likely to be related to the poor vision phenotype. By focusing on genes that are convergently lost, we aim at enriching for the most promising candidates. Indeed, the 37 genes that were removed during this filtering step are not enriched in “visual perception ”, which we now show in Figure 1 – source data 9B, which indicates that this filter is helpful. The rationale behind this filter is now better explained in the Methods section “Forward genomics screen”:

“Finally, to obtain candidates that are convergently lost, we required that a gene was lost in at least three of the seven independent low-acuity lineages. This filtering step removed 37 genes and these genes are not enriched in any GO-term (Figure 1 – source data 9B). Of the remaining 31 genes, two genes are no longer contained in the latest Ensembl annotation (version 106) and were therefore dropped. The other 29 genes are presented in Figure 1 and considered in the enrichment analysis (below).”

20 of the 26 top hits vanish when using a cutoff of 2 instead of 1. With such an impact, it is difficult to interpret what exactly is being measured or what the "right" cutoff may be to get the best measure of vision loss. This is exacerbated by the unintuitive nature of visual acuity scores. Unlike the iconic convergence examples, the operational difference between visual acuity scores of 0.5, 1, and 2, or even 2 and 20, is not obvious. If a continuous measure is going to be binarized, there should be a more apparent argument for the cutoffs, particularly since the 0.5 cutoff has more significant results than 1, but is not the focus of the manuscript.

We agree with the reviewer that it is not trivial to pick a ‘right’ cutoff for visual acuity. We present the results with a cutoff of 1 as this separates the echolocating bats from the non-echolocating Pteropodid bats, which have larger eyes and rely more on vision. However, the ‘right’ cutoff likely does not exist. For this reason, we ran our screen three times, testing not only a cutoff of 1, but also the more inclusive cutoff 1.5 and the more strict cutoff of 0.5. Importantly, SERPINE3 is a significant hit for all three cutoffs, indicating that this gene tends to be lost in species with lower acuity values, irrespective of the exact cutoff.

Given the difficulty of finding a ‘right’ cutoff, we now also investigated an additional phenotype definition that is independent of visual acuity. To this end, we considered a molecular loss signature of known vision genes and defined 10 “poor vision” species that have lost ≥5 known “visual perception” genes. It is reassuring that these 10 species are similar to low-acuity species defined by visual acuity values <1 and only two species differ. Supporting a robust association for *SERPINE3*, this screen also ranked *SERPINE3* among the top hits.

We added the details to the Methods section:

“In addition to using phenotype definitions based on visual acuity values, we also tested a phenotype definition based on a molecular signature of losses of known vision-related genes. To this end, we ranked all 49 species based on their number of lost genes annotated with the GO-term “visual perception” (total of 93 genes, Figure 1 – source data 1). The set of ten species that have lost five or more visual perception genes is similar to the set of low-acuity species used in the original screen. The differences are that mouse and deer mouse are substituted by manatee and goat. Re-running the screen based on this molecular loss pattern of vision genes, retrieved *SERPINE3* at rank 5 (Figure 1 – source data 7).”

and summarized it in the main text:

“To show that *SERPINE3* loss is robustly associated with poor vision, we varied the visual acuity thresholds used to classify species as having low-acuity vision and explored an alternative approach for defining poor vision species based on a molecular loss signature of known vision-related genes. All three modified screens consistently retrieved *SERPINE3* as one of the top-ranked hits (Figure 1 – source data 5-7), showing that this association is robust to the selected thresholds and phenotype definition.”

Filters 3 and 4 appear to be redundant with each other and convergence analysis itself. The analysis is already scoring genes for intactness in conjunction with the trait, so it seems unnecessary to exclude genes from the analysis based on patterns of intactness in conjunction with the trait. It's the same measurement, twice.

The reviewer is right that filter 3 was not really necessary. As explained above, we have now removed this filter, which only affected three genes.

The purpose of these filters is unclear. The consequences of these filters on p-value distributions, multiple hypothesis corrections, and top hits are unclear, and possibly inappropriate. The lack of a negative control again exacerbates these concerns. While it is non-trivial to select a control set (or many sets) of lineages with comparable topological relationships as the experimental group, it is necessary to demonstrate the robustness of the method and dataset.

As mentioned above, we have now explained the rationale behind the filters. We also ran a negative control screen. In contrast to the real screen for genes preferentially lost in low-acuity species, this control screen did not retrieve any eye-related genes. The results of the control screen are provided in Figure 1 – source data 4.

Including entire gene lists in the supplemental tables for each method (as opposed to just the top 26 of a single method) would helpfully provide the raw p-values for the distributions mentioned above, allow more comparison and quantification of differences between cutoffs, and allow for future intersections with other datasets. This would dramatically increase the utility of the overall convergence results for the field. In conjunction, there is a truly impressive amount of genomic work here that is unlikely to be replicated by someone else that wants to do a similar analysis on a different phenotype, but the dataset itself does not seem to be available.

We apologize for not listing the entire dataset in the submitted version. This is now corrected by providing a new source Data file, Figure 1 – source data 2, which lists all genes that were screened, the raw P-values and FDR values, and which genes passed the filter for losses in at least three independent lineages.

Figure 1 – source data 4-7 now provides the full list of genes together with their P-values, FDRs and filters for all additional screens conducted in this manuscript (two different visual acuity cutoffs, using a gene loss signature to define the phenotype, negative control screen).